# Unified Probabilistic Deep Continual Learning through Generative Replay and Open Set Recognition

## Abstract

We introduce a unified probabilistic approach for deep continual learning with open set recognition, based on variational Bayesian inference. Our model combines a joint probabilistic encoder with a generative model and a linear classifier that get shared across tasks. The proposed open set recognition bounds the approximate posterior by fitting regions of high density on the basis of correctly classified data points and balances open set detection with recognition errors. Catastrophic forgetting is significantly alleviated through generative replay, where the open set recognition is used to reject statistical outliers from low density areas of the class specific posterior. Our approach naturally allows for forward and backward transfer while maintaining past knowledge without the necessity of storing old data, regularization or inferring task labels. We demonstrate compelling results in the challenging continual learning scenario of incrementally expanding the single-head classifier for both class incremental visual and audio classification tasks, as well as incremental learning of datasets across modalities. We further quantitatively demonstrate our models ability to successfully distinguish unseen unknown data from trained known tasks and subsequently prevent misclassification.

## 1 Introduction

Most machine learning systems make the closed world assumption and are predominantly trained according to the isolated learning paradigm, where data is available at all times and is independently and identically distributed. However, in the context of continual learning, where tasks and data arrive in sequence, neither of these two principles is desirable. A neural network that is trained exclusively on a new task's data forgets past knowledge and suffers from an early identified phenomenon commonly referred to as catastrophic forgetting (McCloskey & Cohen, 1989). Moreover, to overcome the closed world assumption, inclusion of a "background" class is veritably insufficient as it is impossible to include all unseen concepts and classes explicitly in the loss function beforehand. Likewise, commonly applied thresholding of prediction values doesn't prevent resulting large confidences for unseen classes if the data is far away from any known data (Matan et al., 1990).

Most of the existing continual learning literature concentrates efforts on either alleviating catastrophic forgetting, maximizing knowledge transfer or addressing ways in which to efficiently store subsets of past data. These works have identified weight regularization (McCloskey & Cohen, 1989; Zenke et al., 2017; Kirkpatrick et al., 2017; Li & Hoiem, 2016; Nguyen et al., 2018) and rehearsal techniques (Ratcliff, 1990; Lopez-Paz & Ranzato, 2017; Rebuffi et al., 2017; Bachem et al., 2015) or have postulated methods based on complementary learning systems theory (O'Reilly & Norman, 2003) through dual-model with generative memory approaches (Gepperth & Karaoguz, 2016; Shin et al., 2017; Wu et al., 2018; Farquhar & Gal, 2018; Achille et al., 2018) as mechanisms against catastrophic inference. On the one hand, regularization techniques can work well in principle, but come with the caveat of relying on a new task's proximity to previous knowledge. On the other hand, training and storing separate models, including generative models for generative rehearsal, comes at increased memory cost and doesn't allow for full knowledge sharing, particularly to already stored models. Specifically, the transfer of already attained knowledge to benefit new tasks, known as forward transfer, as well as the potential positive impact of learning new concepts to aid in existing tasks, known as backward transfer, are crucial to any continual learning system. Generally speak-

ing, most current approaches include a set of simplifications, such as considering separate classifiers for each new task, referred to as multi-head classifiers. This scenario prevents "cross-talk" between classifier units by not sharing them, which would otherwise rapidly decay the accuracy (Zenke et al., 2017; Kirkpatrick et al., 2017; Rusu et al., 2016; Shin et al., 2017; Gepperth & Karaoguz, 2016; Rebuffi et al., 2017; Achille et al., 2018; Nguyen et al., 2018) as newly introduced classes directly impact and confuse existing concepts. In the multi-head scenario task ids thus need to be encoded or are often assumed to be given by humans in order to know which classifier to use for prediction. Correspondingly, in generative replay, generative and discriminative models are taken to be separate models (Shin et al., 2017; Farquhar & Gal, 2018; Nguyen et al., 2018). Similar to regularization of a classifier, a generative model can suffer from the learned approximate posterior distribution deviating further from the true posterior with each further task increment. In order to avoid catastrophic forgetting induced by learning to generate on previously generated data, previous works even store a separate generative model per task (Farquhar & Gal, 2018), in analogy to the multi-head classifier. An extended review of recent continual learning methods is provided by Parisi et al. (2019).

A parallel thread pursues a complementary component of identifying out-of-distribution and open set examples. While current continual learning approaches typically do not include this thread yet, it can be considered crucial to any system and a necessity in order to avoid encoding task labels and distinguishing seen from unknown data. Here, multiple methods rely on using confidence values as means of rejection through calibration (Liang et al., 2018; Lee et al., 2018b;a). Arguably this also includes Bayesian approaches using variational methods (Farquhar & Gal, 2018; Achille et al., 2018) or Monte-Carlo dropout (Gal & Ghahramani, 2015) to estimate uncertainties. Since the closed world assumption also holds true for Bayesian methods as the approximated posterior probability cannot be computed for unknown classes, misclassification still occurs, as the open space risk is unbounded (Boult et al., 2019). Recently Thomas et al. (2014); Bendale & Boult (2016); Dhamija et al. (2018) have proposed extreme value theory (EVT) based open set recognition to bound the open-space risk and balance it with recognition errors in deep neural networks.

In this work we propose a probabilistic approach to unify open set recognition with continual learning in a single deep model in order to remove or alleviate above mentioned common simplifications. Specifically, our contributions are:

- We introduce a single model for continual learning that combines a joint probabilistic encoder with a generative model and a linear classifier. Inspired by EVT based open set recognition (Bendale & Boult, 2016) for Softmax prediction layers, this model architecture gives rise to a natural way of open set recognition with statistical outlier rejection on the basis of the approximate posterior in Bayesian inference. The latter requires no upfront knowledge of open set data or corresponding modifications to loss or training procedure and can successfully prevent nonsensical predictions for unseen unknown data, a robustness feature that is currently not present in closed world continual learning systems.

- We show how this EVT bound to the posterior can be used for both identification and rejection of statistically outlying unseen unknown data instances, as well as exclusion of generated samples from areas of low probability density. When used in generative replay this leads to significantly reduced catastrophic forgetting without storing real data.

- We share our model across tasks and automatically expand a single linear classifier head with units for new classes, thus not requiring explicit task labels during inference.

- We demonstrate that our approach can incrementally learn the classes of two image and one audio dataset, as well as cross-dataset scenarios across modalities, while allowing for forward and backward transfer due to weight-sharing. When presented with novel data our model is able to distinguish between unseen data from various datasets and data belonging to known tasks. We further show that our approach readily profits from recent model advances such as variational lossy auto-encoders (Gulrajani et al., 2017; Chen et al., 2017).

## 2 UNIFYING CONTINUAL LEARNING WITH OPEN SET RECOGNITION

In isolated supervised machine learning the core assumption is the presence of i.i.d. data at all times and training is conducted using a dataset $\boldsymbol{D} \equiv \left\{ \left( \boldsymbol{x}^{(n)}, y^{(n)} \right) \right\}_{n=1}^{N}$, consisting of $N$ pairs of data instances $\boldsymbol{x}^{(n)}$ and their corresponding labels $y^{(n)} \in \{1 \ldots C\}$ for $C$ classes. In contrast, in

continual learning task data $\boldsymbol{D}_t \equiv \left\{ \left( \boldsymbol{x}_t^{(n)}, y_t^{(n)} \right) \right\}_{n=1}^{N_t}$ with $t = 1, \ldots, T$ arrives sequentially for $T$ disjoint datasets, each with number of classes $C_t$. In our work, we consider this class incremental scenario from a perspective of variational Bayesian inference in deep neural networks (Kingma & Welling, 2013) that consist of a shared encoder with variational parameters $\boldsymbol{\theta}$, decoder and linear classifier with respective parameters $\phi$ and $\boldsymbol{\xi}$. The joint probabilistic encoder learns an encoding to a latent variable $\boldsymbol{z}$, over which a prior is placed, say a unit Gaussian. Using variational inference, its purpose is to approximate the true posterior to both $p_\phi(\boldsymbol{x}, \boldsymbol{z})$ and $p_{\boldsymbol{\xi}}(y, \boldsymbol{z})$. The probabilistic decoder $p_\phi(\boldsymbol{x}|\boldsymbol{z})$ and probabilistic linear classifier $p_{\boldsymbol{\xi}}(y|\boldsymbol{z})$ then return the conditional probability density of the input $\boldsymbol{x}$ and target $y$ under the respective generative model given a sample $\boldsymbol{z}$ from the approximate posterior $q_{\boldsymbol{\theta}}(\boldsymbol{z}|\boldsymbol{x})$. This yields a generative model $p(\boldsymbol{x}, \boldsymbol{y}, \boldsymbol{z})$, for which we assume a factorization and generative process of the form $p(\boldsymbol{x}, \boldsymbol{y}, \boldsymbol{z}) = p(\boldsymbol{x}|\boldsymbol{z})p(\boldsymbol{y}|\boldsymbol{z})p(\boldsymbol{z})$. For variational inference with our model the following continual learning loss function thus needs to be optimized:

$$
\mathcal{L}_t^{\text{UB}}(\boldsymbol{\theta}, \phi, \boldsymbol{\xi}) = \sum_{\tau=1}^{t} \sum_{n=1}^{N_\tau} [\mathbb{E}_{q_{\boldsymbol{\theta},t}(\boldsymbol{z}|\boldsymbol{x}_\tau^{(n)})}[\log p_{\phi,t}(\boldsymbol{x}_\tau^{(n)}|\boldsymbol{z}) + \log p_{\boldsymbol{\xi},t}(y_\tau^{(n)}|\boldsymbol{z})]
$$
$$
- \beta KL(q_{\boldsymbol{\theta},t}(\boldsymbol{z}|\boldsymbol{x}_\tau^{(n)}) \,||\, p(\boldsymbol{z}))]
\tag{1}
$$

Here, we have added a weight term $\beta$ to the KL divergence to balance the individual loss terms, similar to the work of Zhou et al. (2012) and Higgins et al. (2017). This factor regulates the trade-off between the additional constraint imposed by the classifier, needing to be able to linearly separate the classes given $\boldsymbol{z}$, and the quality of the approximation to the training data. To balance this trade-off irrespective of input and latent dimensionality and number of classes, the losses are normalized according to dimensions. Note that in practice this changes the relative scale of the losses and thus the interpretation of specific $\beta$ values with respect to the original authors Higgins et al. (2017). We provide a more detailed discussion with empirical examples for the role of $\beta$ in the supplementary section. However, equation 1 requires the presence of all data for all tasks and is thus generally not feasible for continual learning where only the most recent task's data is assumed to be available. In context of variational inference, two potential approaches offer solutions to this challenge: a prior-based approach using the former approximate posterior $q_{\boldsymbol{\theta},t-1}$ as the new task's prior (Nguyen et al., 2018) or estimating the likelihood of former data through generative replay or other forms of rehearsal (Farquhar & Gal, 2018; Achille et al., 2018). For our proposed model, we follow the latter line of work and let the prior remain the same at all times. Making use of the generative nature of our model, we let the above upper-bound to task incremental continual learning become:

$$
\mathcal{L}_t(\boldsymbol{\theta}, \phi, \boldsymbol{\xi}) = \sum_{n=1}^{\tilde{N}_t} [\mathbb{E}_{q_{\boldsymbol{\theta},t}(\boldsymbol{z}|\tilde{\boldsymbol{x}}_t^{(n)})}[\log p_{\phi,t}(\tilde{\boldsymbol{x}}_t^{(n)}|\boldsymbol{z}) + \log p_{\boldsymbol{\xi},t}(\tilde{y}_t^{(n)}|\boldsymbol{z})] - \beta KL(q_{\boldsymbol{\theta},t}(\boldsymbol{z}|\tilde{\boldsymbol{x}}_t^{(n)}) \,||\, p(\boldsymbol{z}))]
$$
$$
+ \sum_{n=1}^{N_t} [\mathbb{E}_{q_{\boldsymbol{\theta},t}(\boldsymbol{z}|\boldsymbol{x}_t^{(n)})}[\log p_{\phi,t}(\boldsymbol{x}_t^{(n)}|\boldsymbol{z}) + \log p_{\boldsymbol{\xi},t}(y_t^{(n)}|\boldsymbol{z})] - \beta KL(q_{\boldsymbol{\theta},t}(\boldsymbol{z}|\boldsymbol{x}_t^{(n)}) \,||\, p(\boldsymbol{z}))]
\tag{2}
$$

Here, $\tilde{\boldsymbol{x}}_t \sim p_{\phi,t-1}(\boldsymbol{x}|\boldsymbol{z})$ and $\tilde{y}_t \sim p_{\boldsymbol{\xi},t-1}(y|\boldsymbol{z})$ with $\boldsymbol{z} \sim p(\boldsymbol{z})$ is a sample from the generative model with the corresponding label obtained from the classifier. $\tilde{N}_t$ is the number of total data instances of all previously seen tasks or alternatively a hyper-parameter. This way the expectation of the log-likelihood for all previously seen tasks is estimated and the dataset at any point in time $\tilde{\boldsymbol{D}}_t \equiv (\boldsymbol{x}_t \cup \tilde{\boldsymbol{x}}_t, y_t \cup \tilde{y}_t)$ is a combination of generations from seen past data distributions and the current task's real data. For each newly arriving task with novel labels, the classifier is expanded with newly initialized units. We note that whereas the loss function with generative replay in equation 2 is used for continual training, equation 1 and thus real data is always used for testing.

The model is further trained in a denoising fashion, where noise is added to each input $\boldsymbol{x}$ to avoid over-fitting. This is preferable to weight regularization as it doesn't entail unrecoverable units that are needed to encode later stage concepts. We have accordingly coined our model Classifying Denoising Variational Auto-Encoder (CDVAE). We optionally enhance the probabilistic decoder with an autoregressive variant, where generation of a pixel's value is spatially conditioned on previous pixels (van den Oord et al., 2016; Gulrajani et al., 2017; Chen et al., 2017). Here, the denoising plays an additional crucial role of de-quantization.

Nonetheless, similar to existing dual-model approaches (Shin et al., 2017; Wu et al., 2018; Farquhar & Gal, 2018), by itself both CDVAE and PixCDVAE models accumulate errors as with each iteration

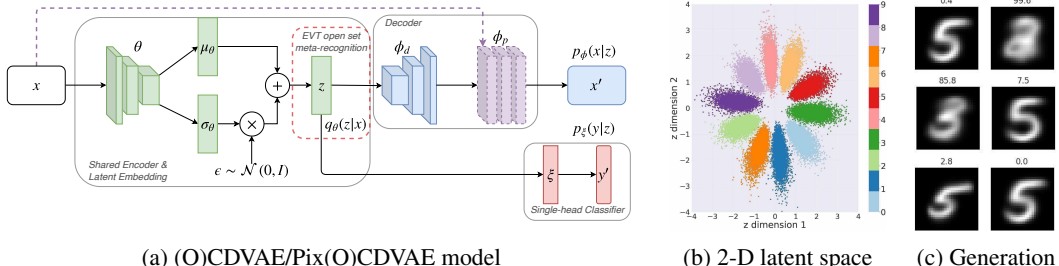

(a) (O)CDVAE/Pix(O)CDVAE model      (b) 2-D latent space    (c) Generation

Figure 1: (a) Joint continual learning model consisting of a shared probabilistic encoder with variational approximation $q_{\boldsymbol{\theta}}(\boldsymbol{z}|\boldsymbol{x})$, probabilistic decoder $p_{\boldsymbol{\phi}}(\boldsymbol{x}, \boldsymbol{z})$ and probabilistic classifier $p_{\boldsymbol{\xi}}(y, \boldsymbol{z})$. The dashed (purple) line denotes an optional pixel decoder with parameters $\boldsymbol{\phi}_p$. For open set recognition and generative replay with outlier rejection, EVT based bounds for the variational approximation are established. (b) 2-D latent space visualization for continually learned incremental MNIST. (c) Generated MNIST images $x \sim p_{\boldsymbol{\phi},t}(\boldsymbol{x}|\boldsymbol{z})$ with $\boldsymbol{z} \sim p(\boldsymbol{z})$ and their corresponding class $c$ obtained from the classifier $p_{\boldsymbol{\xi},t}(y|\boldsymbol{z})$ for $c = 5$, together with their open set outlier probability $\omega_{c,t}$.

of generative replay deviations of the approximate from the true posterior get amplified. However, in our joint model, the linear classifier directly affects the partitioning of the latent space by influencing the joint probabilistic encoder's weights, resulting in class specific areas of large probability density. This is particularly noticeable for lossy VAEs (Gulrajani et al., 2017; Chen et al., 2017) that leave the encoding of local structure to autoregressive layers and hence in our case attribute more influence on the latent space to the classifier. We note that these class specific areas in latent space are not necessarily encouraged for deeper classifiers, however would argue that with a sufficiently expressive probabilistic encoder such a classifier is not necessary. For visualization purposes, we have trained a CDVAE following the details of section 3 with a two-dimensional latent space on the class-incremental MNIST (LeCun et al., 1998) upper-bound and show the latent space embedding for the validation dataset at the end of continual learning in figure 1b. Corresponding intermediate visualizations for each task increment and PixCDVAE can be found in the supplementary material. We take advantage of the classifier's impact on the latent space as the foundation for posterior based open set recognition and complementary generative replay with statistical outlier rejection. We refer to this extended model as Open-set Classifying Denoising Variational Auto-Encoder (OCDVAE) and PixOCDVAE respectively. An illustration of our joint probabilistic model is shown in figure 1a.

## 2.1 OPEN SET RECOGNITION WITH BOUNDS TO THE CLASS SPECIFIC POSTERIOR

We leverage the single-headed linear classifier's presence and the resulting formation of class specific high density regions in latent space as the basis for open set recognition. Specifically, we draw inspiration from the EVT based OpenMax approach (Bendale & Boult, 2016), that uses knowledge about extreme distance values to modify a Softmax prediction's confidence, but propose to instead

---

**Algorithm 1 Open set recognition calibration for deep variational neural networks**. At the end of task $t$, a Weibull model fit of tail-size $\eta$ is conducted to bound the per class approximate posterior. Per class $c$ Weibull models $\boldsymbol{\rho}_{c,t}$ with their respective shift $\tau_{c,t}$, shape $\kappa_{c,t}$ and scale $\lambda_{c,t}$ parameters are returned. The CDVAE model can now be referred to as OCDVAE.

**Require:** CDVAE with probabilistic encoder $q_{\boldsymbol{\theta},t}(\boldsymbol{z}|\boldsymbol{x})$ and probabilistic classifier $p_{\boldsymbol{\xi},t}(y|\boldsymbol{z})$
**Require:** Classifier probabilities $p_{\boldsymbol{\xi},t}(y|\boldsymbol{z})$ and samples from the approximate posterior $\boldsymbol{z}(\boldsymbol{x}^{(i)}) \sim$
     $q_{\boldsymbol{\theta},t}(\boldsymbol{z}|\boldsymbol{x}^{(i)})$ for each training dataset example $\boldsymbol{x}^{(i)}$ in dataset $\tilde{\boldsymbol{D}}_t$
**Require:** For each class $c$, let $\boldsymbol{S}_c^{(i)} = \boldsymbol{z}(\boldsymbol{x}_c'^{(i)})$ for each correctly classified training example $\boldsymbol{x}_c'^{(i)}$
   1: **for** $c = 1 \ldots C$ **do**
   2:      **Compute per class latent mean** $\bar{\boldsymbol{S}}_{c,t} = mean(\boldsymbol{S}_c^{(i)})$
   3:      **Weibull model** $\boldsymbol{\rho}_{c,t} = (\tau_{c,t}, \kappa_{c,t}, \lambda_{c,t}) = $ Fit Weibull $\left(||\boldsymbol{S}_c - \bar{\boldsymbol{S}}_{c,t}||, \eta\right)$
   4: **Return** means $\bar{\boldsymbol{S}}_t$ and Weibull models $\boldsymbol{\rho}_t$

---

---

**Algorithm 2 Open set probability estimation for unknown and uncertain inputs.** At the end of any task $t$, novel data points are considered statistical outliers if a Weibull model's cumulative distribution function's (CDF) outlier probability value exceeds a prior $\Omega_t$.

---

**Require:** OCDVAE with probabilistic encoder $q_{\boldsymbol{\theta},t}(\boldsymbol{z}|\boldsymbol{x})$
**Require:** Per class latent mean $\bar{\boldsymbol{S}}_{c,t}$ and Weibull model $\boldsymbol{\rho}_{c,t}$, each with parameters $(\tau_{c,t}, \kappa_{c,t}, \lambda_{c,t})$
     **For a novel input example** $\hat{\boldsymbol{x}}$ **sample** $\boldsymbol{z} \sim q_{\boldsymbol{\theta},t}(\boldsymbol{z}|\hat{\boldsymbol{x}})$
2:   **Compute distances to** $\bar{\boldsymbol{S}}_{c,t}$: $d_{c,t} = ||\bar{\boldsymbol{S}}_{c,t} - \boldsymbol{z}||$
     **for** $c = 1 \ldots C$ **do**
4:        **Compute Weibull CDF** $\omega_{c,t}(d_{c,t}) = 1 - \exp\left(-\frac{||d_{c,t} - \tau_{c,t}||}{\lambda_{c,t}}\right)^{\kappa_{c,t}}$
     **Reject input** if $\omega_{c,t}(d_{c,t}) > \Omega_t$ for any class $c$.

---

bound the open-space risk by employing statistical outlier rejection on the basis of the approximate posterior in Bayesian inference. Considering a trained model at the end of task $t$, the EVT based open set recognition fits a Weibull distribution on the distances of each correctly classified training example's sample from the approximate posterior $\boldsymbol{z}(\boldsymbol{x}) \sim q_{\boldsymbol{\theta},t}(\boldsymbol{z}|\boldsymbol{x})$ to the respective per class sample mean. In other words, regions of high density of the approximate posterior for each class are identified for the subset of correctly identified data points, with the tail of the Weibull distribution bounding the open-space as well as regions of low-density. This procedure is described in algorithm 1. Once these bounds are established, for any novel input, the Weibull models' cumulative distribution function can be used to estimate the statistical outlier probability, based on the unknown example's sample(s) from the posterior and their distance to the class' region of highest density. If the outlier probability is larger than a prior rejection probability, the input can be considered as unknown and a false overconfident classifier prediction avoided, or conversely it is classified into the already existing classes across all known tasks as detailed in algorithm 2.

## 2.2 GENERATIVE REPLAY WITH STATISTICAL OUTLIER REJECTION

As the obtained open set recognition models provide bounds between the posterior's regions of high and low density, we can extend the use from rejection of statistical outliers for novel input examples to rejection of samples drawn directly from the prior for the purpose of generative replay. Consider generation of a data point $\boldsymbol{x} \sim p_{\boldsymbol{\phi},t}(\boldsymbol{x}|\boldsymbol{z})$. It is common practice to assume that the approximated posterior is close to the true posterior. If a sample from the prior $\boldsymbol{z} \sim p(\boldsymbol{z})$ stems from an area of low density, one further inherently relies on the generative model's capability for interpolation. In periodic generative rehearsal, these factors can entail accumulation of errors through increasing deviations between approximated and true posterior, as well as classifier confusion due to ambiguous examples. To inhibit the latter and as a result implicitly the former, our obtained bounds can be

---

**Algorithm 3 Generative replay with outlier rejection.** For generative replay after training task $t$, samples $\boldsymbol{z} \sim p(\boldsymbol{z})$ are rejected if the Weibull CDF's probability value exceeds the prior $\Omega_t$.

---

**Require:** OCDVAE with probabilistic encoder $q_{\boldsymbol{\theta},t}(\boldsymbol{z}|\boldsymbol{x})$ and probabilistic classifier $p_{\boldsymbol{\xi},t}(y|\boldsymbol{z})$
**Require:** Per class latent mean $\bar{\boldsymbol{S}}_{c,t}$ and Weibull model $\boldsymbol{\rho}_{c,t}$, each with parameters $(\tau_{c,t}, \kappa_{c,t}, \lambda_{c,t})$
**Require:** Number of samples per class $M_c \forall c = 1, \ldots, \tilde{C}_t$ with $\tilde{C}_t$ seen classes up to task $t$
     **Initialize:** $m_c \leftarrow 0 \forall c = 1, \ldots, \tilde{C}$, $\tilde{\boldsymbol{X}}_t = \emptyset$ and $\tilde{\boldsymbol{Y}}_t = \emptyset$
     **while** $\sum_{c=1}^{\tilde{C}} m_c < \sum_{c=1}^{\tilde{C}} M_c$ **do**                        ▷ in parallel
3:   **Sample from prior** $\boldsymbol{z} \sim p(\boldsymbol{z})$
     **Compute label** $\hat{c} = \texttt{argmax}\left(\log p_{\boldsymbol{\xi},t}(y|\boldsymbol{z})\right)$
     **Calculate distance** $d_{\hat{c},t} = ||\bar{\boldsymbol{S}}_{\hat{c},t} - \boldsymbol{z}||$
6:   **Compute Weibull CDF** $\omega_{\hat{c},t}(d_{\hat{c},t}) = 1 - \exp\left(-\frac{||d_{\hat{c},t} - \tau_{\hat{c},t}||}{\lambda_{\hat{c},t}}\right)^{\kappa_{z_{\hat{c},t}}}$
     **if** $\omega_{\hat{c},t} < \Omega_t$ and $m_{\hat{c}} < M_{\hat{c}}$ **then**
         **Calculate decoder** $\tilde{\boldsymbol{x}} \sim p_{\boldsymbol{\phi},t}(\boldsymbol{x}|\boldsymbol{z})$
9:        **Append to dataset** $\tilde{\boldsymbol{X}}_t \leftarrow \tilde{\boldsymbol{X}}_t \cup \tilde{\boldsymbol{x}}$ and $\tilde{\boldsymbol{Y}}_t \leftarrow \tilde{\boldsymbol{Y}}_t \cup \hat{c}$ and $m_{\hat{c}} \leftarrow m_{\hat{c}} + 1$
     **else**    reject

---

exploited by rejecting samples from low density regions and replacing them with statistically inlying samples. Hence, we extend generative replay for the OCDVAE with such a rejection mechanism. We now first sample from the prior until a desired amount of statistical inliers per class is reached, whereas the label is obtained using the linear classifier and is accepted if it is in correspondence with the respective class' Weibull model. We then proceed to generate the dataset with the probabilistic decoder. This bounded version of generative replay with statistical outlier rejection is detailed in algorithm 3. An example of MNIST images with outlier probabilities based on their sample from the prior are shown in figure 1c to illustrate the rejection of ambiguous and misclassified instances, with additional images in the supplementary material. The reason we use sampling with rejection is because our Weibull models are based on scalar distances and thus samples from the Weibull distributions cannot be inverted to high-dimensional $z$ vectors. While this may sound detrimental to our method, we argue that both sampling from the prior $z \sim p(z)$ in large parallelized batches and likewise computation of a single layer classifier, even in high dimensions, is computationally negligible in contrast to the much more computationally heavy deep probabilistic decoder. The latter further only needs to be processed for accepted samples and thus does not add any computational complexity with respect to conventional generative replay. Note that the amount of samples that has to be drawn before a sample is accepted scales with the dimensionality of the latent space and could in principle be regarded as a limitation for very high dimensional latent spaces. However, practical VAEs typically do not tend to profit from very high dimensional latent spaces.

## 3 Experiments

Similar to recent literature (Zenke et al., 2017; Kirkpatrick et al., 2017; Farquhar & Gal, 2018; Shin et al., 2017; Parisi et al., 2019), we consider the incremental MNIST (LeCun et al., 1998) dataset, where classes arrive in groups of two, and corresponding versions of the FashionMNIST (Xiao et al., 2017) and AudioMNIST dataset (Becker et al., 2018). For the latter we follow the authors' procedure of converting the audio recordings into spectrograms, resized to $32 \times 32$. In addition to this class incremental setting, we evaluate cross-dataset scenarios with all inputs resized to $32 \times 32$, where dataset are sequentially added with all of their classes and the model has to learn across modalities.

We compare our proposed OCDVAE model with its counterpart CDVAE to highlight the improvement induced by algorithm 3. We further contrast these improvements with the dual model variant, consisting of a VAE for generative replay and a separate deep model for classification (Shin et al., 2017). Further, autoregressive pixel model variants are reported to demonstrate how approaches benefit from more recent advances in model architecture. We evaluate elastic weight consolidation (EWC) (Kirkpatrick et al., 2017) on the classification task without a decoder to show that approaches based on regularization fail at maintaining previous knowledge in a single-head classifier scenario. Although the latter has already been shown in a recent review by Parisi et al. (2019) and even for multi-head classifier scenarios (Kemker et al., 2018), we nevertheless provide these results for emphasis. We do not report episodic memory approaches like coresets (Bachem et al., 2015) that explicitly store real data or additional regularization as suggested by Nguyen et al. (2018). These methods are complementary to our proposed approach, reporting them separately might mislead the reader and an evaluation that additionally includes these methods on top is left for future work. To provide a frame of reference for achievable performance, we further consider upper- and lower-bounds for our joint model. The CDVAE lower-bound is obtained when only the current task's data is available and provides the worst case performance where absolute catastrophic forgetting occurs. Conversely, the upper-bound is obtained with equation 1 when a task's data is added to all previous tasks' real data and yields a model's maximum achievable performance if trained in an incremental fashion. The isolated learning baseline corresponds to the typical machine learning practice outside of continual learning where all tasks' data is always present. All models can be trained on a single GTX 1080 GPU and we will make our code publicly available.

### 3.1 Metrics

Our metrics are inspired by previously proposed continual learning classification measures (Lopez-Paz & Ranzato, 2017; Kemker et al., 2018). In addition to overall accuracy, these metrics monitor forgetting by computing a base accuracy on the initial task, while also gauging the amount of new knowledge that can be encoded by monitoring the accuracy for the most recent increment. In the

multi-head classification scenario, both the overall and base accuracy is then divided with an ideal accuracy. As our single-head classifier scenario implies a natural decay of a task's base accuracy with increasing amount of classes, we instead report the raw base and new accuracies and compare them with the upper-bound and isolated performance. This is a particularly important distinction to the multi-head scenario reported in many previous works, where the amount of forgetting can easily be determined by the amount that the accuracy of each binary classifier decays over time as, introduction of new classes does not directly affect the other tasks' weights. We extend these concepts to the probabilistic decoder's reconstruction loss. Our metrics are thus:

- **Classification accuracy**: base accuracy $\alpha_{t,base}$ of initial task at increment $t$. New accuracy $\alpha_{t,new}$ for the freshly added task. Accuracy $\alpha_{t,all}$ over all classes of all tasks seen so far.

- **Reconstruction negative-log-likelihood (NLL)**: base NLL $\gamma_{t,base}$ of initial task at task increment $t$. New NLL $\gamma_{t,new}$ for the added task. NLL $\gamma_{t,all}$ for all tasks seen so far.

- **Kullback-Leibler Divergence**: between the approximate posterior $q_{\boldsymbol{\theta},t}(\boldsymbol{z}|\boldsymbol{x})$ and the prior $p(\boldsymbol{z})$ distribution and thus always evaluated for all data up to and including task $t$.

## 3.2 TRAINING HYPER-PARAMETERS

We base our encoder and decoder architecture on 14-layer wide residual networks (He et al., 2016; Zagoruyko & Komodakis, 2016) as used in lossy auto-encoders (Gulrajani et al., 2017; Chen et al., 2017), with a latent dimensionality of 60 to demonstrate scalability to high-dimensions and deep networks. Our classifier always consists of a single linear layer. The optional autoregressive decoder adds three additional layers. For a common frame of reference, all methods' share the same WRN architecture. We use hyper-parameters consistent with the literature (Gulrajani et al., 2017; Chen et al., 2017). Accordingly, all models are optimized using stochastic gradient descent with a mini-batch size of 128 and Adam (Kingma & Ba, 2015) with a learning rate of 0.001, batch-normalization (Ioffe & Szegedy, 2015) in all hidden layers with a value of $10^{-5}$ and ReLU activations. We add noise sampled from $\mathcal{N}(0, 0.25)$ to the input to avoid over-fitting. Due to the inevitable data augmentation effect, we train all approaches in this denoising fashion. No further data augmentation or preprocessing is applied. We initialize all weights according to He et al. (2015). For our single-head expanding classifier this ensures that all units are always initialized from the same distribution when they get added at each task increment, as the initialization depends only on the layer's input dimensionality. This is not necessarily the case for other weight initialization techniques where a dependency on the amount of classifier units exists. We train all class incremental models for 120 epochs per task on MNIST and FashionMNIST and 150 epochs on AudioMNIST. Complementary incremental cross-dataset models are trained for 200 epochs per task. While our proposed model exhibits forward transfer due to weight sharing and need not necessarily be trained for the entire amount of epochs for each subsequent task, this guarantees convergence and a fair comparison of results. Isolated models are trained for 200 and 300 epochs until convergence respectively. For the generative replay with statistical outlier rejection, we use an aggressive rejection rate of $\Omega_t = 0.01$ (with analogous results with 0.05) and dynamically set tail-sizes to 5% of seen examples per class. The used open set distance measure is the cosine distance. We provide a detailed description of architectures and EWC hyper-parameters in the appendix. Results are averaged over five experimental repetitions, apart from the isolated, lower- and upper-bound that show negligible deviations.

## 3.3 INCREMENTAL CONTINUAL LEARNING RESULTS AND DISCUSSION

Results for the class incremental scenarios for all models, the upper- and lower-bounds and the isolated setting are shown in table 1. Corresponding results for the two directions of incremental cross-dataset experiments are summarized in table 2. In general the upper-bound values are almost identical to isolated learning. Similarly, the new task's metrics are negligibly close, as the WRN architecture ensures enough capacity to encode new knowledge. In contrast to EWC that is universally unable to maintain its old knowledge, CDVAE and PixCDVAE are able to partially retain previous information. Yet they accumulate errors due to samples generated from low density regions. While the dual model approaches do not exhibit this behavior for MNIST, they displays similar forgetting for other experiments, particularly for Audio data. However, our proposed OCDVAE and PixOCD-VAE generative replay overcomes this issue to a considerable degree. For the class incremental scenarios the best models feature less than 10% drop in accuracy on all datasets even with repeated

Table 1: Results for class incremental continual learning approaches averaged over 5 runs, baselines and the reference isolated learning scenario for the three datasets. $\alpha_T$ and $\gamma_T$ indicate the respective accuracy and NLL reconstruction metrics at the end of the last task increment $T = 5$. $KL_T$ denotes the corresponding KL divergence. Arrows indicate whether lower or larger values are better.

| Class-incremental | | $\alpha_T(\%)\uparrow$ | | | $\gamma_T$(nats)$\downarrow$ | | | $KL_T$(nats)$\downarrow$ |
|---|---|---|---|---|---|---|---|---|
| | | base | new | all | base | new | all | all |
| **MNIST** | CDVAE ISO | | | 99.45 | | | 78.12 | 22.12 |
| | CDVAE UB | 99.57 | 99.10 | 99.29 | 64.99 | 85.88 | 81.97 | 21.01 |
| | CDVAE LB | 00.00 | 99.85 | 20.16 | 123.2 | 92.00 | 163.7 | 24.87 |
| | EWC | $00.45_{\pm 0.059}$ | $99.58_{\pm 0.052}$ | $20.26_{\pm 0.027}$ | | | | |
| | Dual Model | $97.31_{\pm 0.489}$ | $98.59_{\pm 0.106}$ | $96.64_{\pm 0.079}$ | $75.08_{\pm 0.623}$ | $89.32_{\pm 0.626}$ | $88.29_{\pm 0.363}$ | $16.13_{\pm 0.225}$ |
| | Dual Pix Model | $98.04_{\pm 1.397}$ | $97.31_{\pm 0.575}$ | $96.52_{\pm 0.658}$ | $92.05_{\pm 1.212}$ | $113.4_{\pm 0.820}$ | $106.1_{\pm 0.868}$ | $9.296_{\pm 1.346}$ |
| | CDVAE | $19.86_{\pm 7.396}$ | $99.00_{\pm 0.100}$ | $64.34_{\pm 4.903}$ | $101.6_{\pm 8.347}$ | $93.55_{\pm 0.391}$ | $107.6_{\pm 1.724}$ | $30.61_{\pm 1.240}$ |
| | PixCDVAE | $56.53_{\pm 4.032}$ | $96.77_{\pm 0.337}$ | $83.61_{\pm 0.927}$ | $102.4_{\pm 6.195}$ | $118.2_{\pm 1.572}$ | $118.7_{\pm 5.320}$ | $16.37_{\pm 0.970}$ |
| | OCDVAE | $92.35_{\pm 4.485}$ | $99.06_{\pm 0.171}$ | $93.24_{\pm 3.742}$ | $77.16_{\pm 1.104}$ | $89.68_{\pm 0.618}$ | $92.92_{\pm 2.283}$ | $21.02_{\pm 0.717}$ |
| | PixOCDVAE | $97.44_{\pm 0.785}$ | $98.63_{\pm 0.430}$ | $96.84_{\pm 0.346}$ | $100.5_{\pm 4.942}$ | $113.3_{\pm 0.755}$ | $111.9_{\pm 2.663}$ | $12.49_{\pm 0.551}$ |
| **FashionMNIST** | CDVAE ISO | | | 89.54 | | | 224.8 | 23.27 |
| | CDVAE UB | 92.20 | 97.50 | 89.24 | 208.4 | 246.2 | 226.2 | 20.27 |
| | CDVAE LB | 00.00 | 99.80 | 19.97 | 306.5 | 242.0 | 275.1 | 21.61 |
| | EWC | $00.17_{\pm 0.076}$ | $99.60_{\pm 0.023}$ | $20.06_{\pm 0.059}$ | | | | |
| | Dual Model | $94.26_{\pm 0.192}$ | $93.55_{\pm 0.708}$ | $63.21_{\pm 1.957}$ | $217.7_{\pm 1.510}$ | $242.8_{\pm 0.898}$ | $230.5_{\pm 1.543}$ | $11.45_{\pm 0.228}$ |
| | Dual Pix Model | $60.04_{\pm 5.151}$ | $98.85_{\pm 0.141}$ | $72.41_{\pm 2.941}$ | $274.1_{\pm 0.349}$ | $305.8_{\pm 0.286}$ | $289.5_{\pm 0.396}$ | $9.781_{\pm 1.068}$ |
| | CDVAE | $39.51_{\pm 7.173}$ | $96.92_{\pm 0.774}$ | $58.82_{\pm 2.521}$ | $232.8_{\pm 5.048}$ | $248.8_{\pm 0.398}$ | $242.2_{\pm 0.754}$ | $26.68_{\pm 0.859}$ |
| | PixCDVAE | $47.83_{\pm 13.41}$ | $97.91_{\pm 0.596}$ | $63.05_{\pm 1.826}$ | $241.1_{\pm 1.747}$ | $283.2_{\pm 2.150}$ | $271.2_{\pm 2.117}$ | $22.14_{\pm 0.377}$ |
| | OCDVAE | $60.63_{\pm 12.16}$ | $96.51_{\pm 0.707}$ | $69.88_{\pm 1.712}$ | $222.8_{\pm 1.632}$ | $244.0_{\pm 0.646}$ | $234.6_{\pm 0.823}$ | $20.47_{\pm 0.742}$ |
| | PixOCDVAE | $74.45_{\pm 2.889}$ | $98.63_{\pm 0.176}$ | $80.85_{\pm 0.721}$ | $234.1_{\pm 1.498}$ | $283.5_{\pm 2.458}$ | $267.2_{\pm 0.586}$ | $17.93_{\pm 0.360}$ |
| **AudioMNIST** | CDVAE ISO | | | 97.75 | | | 429.7 | 17.89 |
| | CDVAE UB | 98.42 | 98.67 | 97.87 | 418.4 | 421.3 | 427.2 | 15.15 |
| | CDVAE LB | 00.00 | 100.0 | 20.02 | 432.9 | 425.2 | 440.4 | 14.52 |
| | EWC | $00.11_{\pm 0.007}$ | $99.41_{\pm 0.207}$ | $19.98_{\pm 0.032}$ | | | | |
| | Dual Model | $61.58_{\pm 0.747}$ | $89.41_{\pm 0.691}$ | $47.42_{\pm 1.447}$ | $425.2_{\pm 0.244}$ | $422.7_{\pm 0.784}$ | $432.7_{\pm 0.385}$ | $5.470_{\pm 0.055}$ |
| | Dual Pix Model | $64.60_{\pm 8.739}$ | $98.18_{\pm 0.885}$ | $75.50_{\pm 3.032}$ | $435.1_{\pm 1.915}$ | $431.9_{\pm 1.032}$ | $440.3_{\pm 1.297}$ | $6.031_{\pm 0.832}$ |
| | CDVAE | $59.36_{\pm 7.147}$ | $84.93_{\pm 6.297}$ | $81.49_{\pm 1.944}$ | $422.7_{\pm 0.182}$ | $423.9_{\pm 0.681}$ | $431.4_{\pm 0.255}$ | $22.96_{\pm 0.912}$ |
| | PixCDVAE | $29.94_{\pm 18.47}$ | $97.00_{\pm 0.520}$ | $63.44_{\pm 5.252}$ | $431.4_{\pm 0.666}$ | $428.0_{\pm 0.851}$ | $436.9_{\pm 0.751}$ | $27.14_{\pm 1.139}$ |
| | OCDVAE | $79.73_{\pm 4.070}$ | $89.52_{\pm 6.586}$ | $87.72_{\pm 1.594}$ | $423.5_{\pm 0.586}$ | $422.9_{\pm 0.537}$ | $430.9_{\pm 0.541}$ | $18.52_{\pm 1.131}$ |
| | PixOCDVAE | $75.25_{\pm 10.18}$ | $99.43_{\pm 0.495}$ | $90.23_{\pm 1.139}$ | $432.3_{\pm 0.189}$ | $429.7_{\pm 1.223}$ | $437.7_{\pm 0.432}$ | $17.45_{\pm 0.835}$ |

Table 2: Results for incremental cross-dataset continual learning approaches averaged over 5 runs, baselines and the reference isolated learning scenario for the three datasets. $\alpha_T$ and $\gamma_T$ indicate the respective accuracy and NLL reconstruction metrics at the end of the last increment $T = 3$. $KL_T$ denotes the corresponding KL divergence. Arrows indicate whether lower or larger values are better.

| Cross-dataset | | $\alpha_T(\%)\uparrow$ | | | $\gamma_T$(nats)$\downarrow$ | | | $KL_T$(nats)$\downarrow$ |
|---|---|---|---|---|---|---|---|---|
| | | base | new | all | base | new | all | all |
| **Fashion-MNIST-Audio** | CDVAE ISO | | | 94.95 | | | 269.6 | 24.97 |
| | CDVAE UB | 89.10 | 97.88 | 95.00 | 311.2 | 434.3 | 269.7 | 25.20 |
| | CDVAE LB | 00.00 | 98.12 | 22.70 | 689.7 | 341.0 | 511.7 | 98.74 |
| | EWC | $22.85_{\pm 0.294}$ | $93.31_{\pm 0.138}$ | $43.42_{\pm 0.063}$ | | | | |
| | Dual Model | $81.89_{\pm 0.104}$ | $96.78_{\pm 0.067}$ | $91.75_{\pm 0.064}$ | $320.0_{\pm 1.275}$ | $431.1_{\pm 1.474}$ | $273.7_{\pm 1.174}$ | $12.80_{\pm 0.060}$ |
| | Dual Pix Model | $82.88_{\pm 0.116}$ | $97.23_{\pm 0.212}$ | $92.16_{\pm 0.061}$ | $288.5_{\pm 0.723}$ | $437.7_{\pm 0.404}$ | $251.6_{\pm 0.231}$ | $9.025_{\pm 1.378}$ |
| | CDVAE | $57.70_{\pm 4.480}$ | $96.73_{\pm 0.235}$ | $81.10_{\pm 1.769}$ | $360.9_{\pm 20.15}$ | $432.1_{\pm 0.231}$ | $296.4_{\pm 7.966}$ | $44.29_{\pm 4.047}$ |
| | PixCDVAE | $56.44_{\pm 1.831}$ | $97.50_{\pm 0.184}$ | $80.76_{\pm 0.842}$ | $289.8_{\pm 1.283}$ | $438.1_{\pm 0.990}$ | $252.6_{\pm 1.424}$ | $29.99_{\pm 0.629}$ |
| | OCDVAE | $80.11_{\pm 2.922}$ | $97.63_{\pm 0.042}$ | $91.13_{\pm 1.045}$ | $345.1_{\pm 7.446}$ | $430.7_{\pm 0.600}$ | $280.2_{\pm 1.069}$ | $25.42_{\pm 1.876}$ |
| | PixOCDVAE | $81.84_{\pm 0.212}$ | $97.75_{\pm 0.169}$ | $91.76_{\pm 0.212}$ | $288.8_{\pm 0.141}$ | $437.1_{\pm 0.725}$ | $251.8_{\pm 0.636}$ | $21.07_{\pm 0.248}$ |
| **Audio-MNIST-Fashion** | CDVAE ISO | | | 94.95 | | | 269.6 | 24.97 |
| | CDVAE UB | 97.17 | 89.16 | 94.91 | 428.8 | 311.9 | 268.2 | 23.91 |
| | CDVAE LB | 00.00 | 89.72 | 34.51 | 506.6 | 311.0 | 351.1 | 34.13 |
| | EWC | $3.420_{\pm 0.026}$ | $87.54_{\pm 0.214}$ | $45.42_{\pm 0.731}$ | | | | |
| | Dual Model | $66.82_{\pm 0.337}$ | $89.15_{\pm 0.050}$ | $87.70_{\pm 0.102}$ | $447.3_{\pm 6.700}$ | $308.5_{\pm 0.599}$ | $270.9_{\pm 1.299}$ | $12.89_{\pm 0.109}$ |
| | Dual Pix Model | $71.58_{\pm 2.536}$ | $88.76_{\pm 0.255}$ | $88.61_{\pm 0.547}$ | $445.8_{\pm 1.601}$ | $290.4_{\pm 0.603}$ | $255.0_{\pm 0.533}$ | $9.164_{\pm 1.312}$ |
| | CDVAE | $79.74_{\pm 2.431}$ | $88.50_{\pm 0.126}$ | $89.46_{\pm 0.600}$ | $448.6_{\pm 5.187}$ | $315.1_{\pm 1.305}$ | $281.6_{\pm 3.205}$ | $33.38_{\pm 0.898}$ |
| | PixCDVAE | $49.38_{\pm 2.256}$ | $88.54_{\pm 0.042}$ | $82.18_{\pm 0.672}$ | $441.4_{\pm 0.495}$ | $287.0_{\pm 0.212}$ | $252.5_{\pm 0.201}$ | $30.60_{\pm 1.556}$ |
| | OCDVAE | $94.53_{\pm 0.283}$ | $89.53_{\pm 0.367}$ | $94.06_{\pm 0.156}$ | $433.4_{\pm 0.424}$ | $311.6_{\pm 0.353}$ | $271.2_{\pm 0.424}$ | $23.16_{\pm 0.121}$ |
| | PixOCDVAE | $91.90_{\pm 0.282}$ | $89.91_{\pm 0.177}$ | $93.82_{\pm 0.354}$ | $438.5_{\pm 1.626}$ | $289.4_{\pm 0.356}$ | $251.3_{\pm 0.354}$ | $20.35_{\pm 0.424}$ |

generative replay. Even stronger results can be observed for the cross-dataset scenarios, where forgetting is alleviated to the extent that final accuracy values are close to the upper bound. Likewise improvements are noticeable in the reconstruction NLL and KL divergences. The OCDVAE models

can consequently achieve reconstruction likelihoods akin to a dual model's separate VAE, while fully sharing encoded knowledge and maintaining a classifier. As a result of OCDVAE's shared weights, we observe backward transfer for some experiments. This is particularly apparent for AudioMNIST, where addition of the second increment first decays and inclusion of later tasks improves the second task's accuracy. Due to space constraints, we provide a detailed account of all intermediate results and examples of generated images for all increments in the supplementary material. We note that the pixel decoders are trained for classification and reported NLL values are obtained through sampling from the multinomial distribution. Original losses are provided in the supplementary material.

### 3.4 OUT OF DISTRIBUTION DETECTION RESULTS AND DISCUSSION

In addition to previous results with focus on continual learning accuracy, where the open set recognition approach has been used for generative replay, we proceed to quantitatively analyze the models' ability to distinguish unknown tasks' data from data belonging to known tasks. Here, the challenge is to consider all unseen test data of already trained tasks as inlying, while successfully recognizing 100 % of unknown datasets as outliers. For this purpose, trained models on each of the three datasets are evaluated on their test set, the remaining datasets and additionally the KMNIST (Clanuwat et al., 2018), SVHN (Netzer et al., 2011), CIFAR10 and CIFAR100 (Krizhevsky, 2009) datasets.

We compare and contrast three criteria that could be used for open set recognition: classifier predictive entropy, reconstruction loss and our proposed latent space based EVT approach. Naively one might expect the Bayesian approach to yield distributions that are sufficiently different for unknown data. We thus evaluate the respective expectation for each criterion with 100 variational samples from the approximate posterior per data point. Figure 2 shows the three criteria and respective percentage of the total dataset being considered as outlying for the OCDVAE model trained on FashionMNIST. In consistence with recent literature (Nalisnick et al., 2019), we can observe that use of reconstruction loss can sometimes distinguish between the known tasks' test data and unknown datasets, but results in failure for others. In the case of classifier predictive entropy, depending on the exact choice of entropy threshold, generally only a partial separation can be achieved. Furthermore, both of these criteria pose the additional challenge of results being highly dependent on the choice of the precise cut-off value. While drawing $z$ samples multiple times, as well as repeatedly calculating the classifier, is computationally feasible, we further note that sampling the entire decoder is computationally prohibitively expensive in practice. In contrast to the other criteria, the test data from the known tasks is regarded as inlying across a wide range of rejection priors $\Omega_t$ and the majority of other datasets is consistently regarded as outlying by our proposed open set mechanism.

As our latent space based EVT approach for open set recognition and respective algorithms 1 and 2 do not technically require a decoder, we can evaluate similar outlier detection for the variational dual model approach. While we also evaluate this scenario, note that the distinction the generative model makes need not necessarily apply to a separate classifier and vice versa. We report the outlier detection accuracy in table 3. Here, a $5\%$ validation split is used to determine the respective value

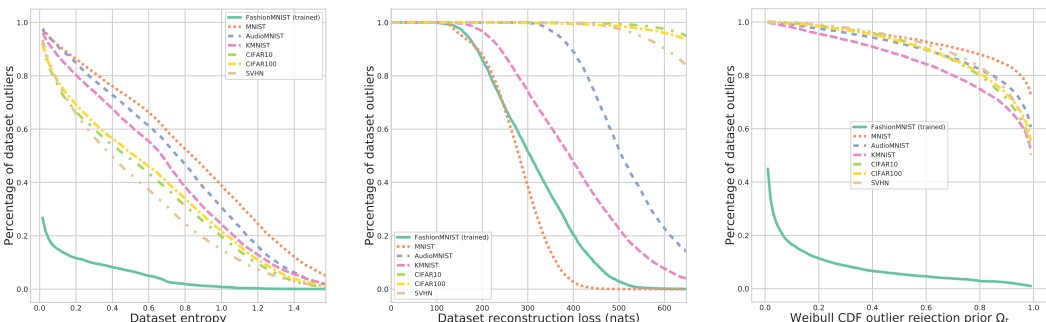

Figure 2: Trained FashionMNIST OCDVAE evaluated on unknown datasets. All metrics are averaged over 100 approximate posterior samples per data point. (Left) Classifier entropy values are insufficient to separate most of unknown from the known task's test data. (Center) Reconstruction loss allows for a partial distinction. (Right) Our posterior-based open set recognition considers the large majority of unknown data as statistical outliers across a wide range of rejection priors $\Omega_t$.

Table 3: Test accuracies and outlier detection values of the joint OCDVAE and dual model approaches when considering 95 % of known tasks' validation data is inlying. Percentage of detected outliers is reported based on classifier predictive entropy, reconstruction loss and our posterior based EVT approach, averaged over 100 $z \sim q_\theta(z|x)$ samples per data-point respectively.

| Outlier detection at 95% validation inliers (%) | | | | MNIST | Fashion | Audio | KMNIST | CIFAR10 | CIFAR100 | SVHN |
|---|---|---|---|---|---|---|---|---|---|---|
| Trained | Model | Test acc. | Criterion | | | | | | | |
| **MNIST** | Dual, CL + VAE | 99.40 | Class entropy | 4.160 | 90.43 | 97.53 | 95.29 | 98.54 | 98.63 | 95.51 |
| | | | Reconstruction | 5.522 | 99.98 | 99.97 | 99.98 | 99.99 | 99.96 | 99.98 |
| | | | Latent EVT | 4.362 | 99.41 | 99.80 | 99.86 | 99.95 | 99.97 | 99.52 |
| | Joint, OCDVAE | 99.53 | Class entropy | 3.948 | 95.15 | 98.55 | 95.49 | 99.47 | 99.34 | 97.98 |
| | | | Reconstruction | 5.083 | 99.50 | 99.98 | 99.91 | 99.97 | 99.99 | 99.98 |
| | | | Latent EVT | 4.361 | 99.78 | 99.67 | 99.73 | 99.96 | 99.93 | 99.70 |
| **FashionMNIST** | Dual, CL + VAE | 90.48 | Class entropy | 74.71 | 5.461 | 69.65 | 77.85 | 24.91 | 28.76 | 36.64 |
| | | | Recon | 5.535 | 5.340 | 64.10 | 31.33 | 99.50 | 98.41 | 97.24 |
| | | | Latent EVT | 96.22 | 5.138 | 93.00 | 91.51 | 71.82 | 72.08 | 73.85 |
| | Joint, OCDVAE | 90.92 | Class Entropy | 66.91 | 5.145 | 61.86 | 56.14 | 43.98 | 46.59 | 37.85 |
| | | | Reconstruction | 0.601 | 5.483 | 63.00 | 28.69 | 99.67 | 98.91 | 98.56 |
| | | | Latent EVT | 96.23 | 5.216 | 94.76 | 96.07 | 96.15 | 95.94 | 96.84 |
| **AudioMNIST** | Dual, CL + VAE | 98.53 | Class entropy | 97.63 | 57.64 | 5.066 | 95.53 | 66.49 | 65.25 | 54.91 |
| | | | Reconstruction | 6.235 | 46.32 | 4.433 | 98.73 | 98.63 | 98.63 | 97.45 |
| | | | Latent EVT | 99.82 | 78.74 | 5.038 | 99.47 | 93.44 | 92.76 | 88.73 |
| | Joint, OCDVAE | 98.57 | Class entropy | 99.23 | 89.33 | 5.731 | 99.15 | 92.31 | 91.06 | 85.77 |
| | | | Reconstruction | 0.614 | 38.50 | 3.966 | 36.05 | 98.62 | 98.54 | 96.99 |
| | | | Latent EVT | 99.91 | 99.53 | 5.089 | 99.81 | 100.0 | 99.99 | 99.98 |

at which $95\%$ of the validation data is considered as inlying before using these priors to determine outlier counts for the known tasks' test set as well as other datasets. While MNIST seems to be a distinct and easy enough dataset for all approaches, we can make two further major observations:

1. The latent based EVT approach outperforms the other criteria, particularly for the OCDVAE where a near perfect open set detection can be achieved.

2. Even though we can apply our proposed open set approach to just a classifier, the joint model with shared latent space consistently exhibits higher outlier detection values. While future investigation into this aspect is needed, we hypothesize that this is due to the joint model also optimizing a variational lower bound to the data distribution $p(x)$.

We provide figures analogous to figure 2 for all models reported in table 3 in the supplementary material. We emphasize that our open set detection does not rely on knowing any open set data examples beforehand as we do not need modification to the loss function or other forms of explicit training. As there exists a body of complementary work (Liang et al., 2018; Lee et al., 2018b; Dhamija et al., 2018) that could readily be integrated, we leave such analysis for future work.

## 4 Conclusion and outlook

We have proposed a unified probabilistic approach to deep continual learning. At the heart lies Bayesian inference with a model combining a shared probabilistic encoder with a generative model and a expanding linear classifier. Weight sharing across tasks allows for forward and backward transfer, while generative replay alleviates catastrophic forgetting. We have then introduced EVT based bounds to the approximate posterior enabled through class specific latent space partitioning induced by the classifier. Derived open set recognition and corresponding generative replay with statistical outlier rejection have been shown to achieve compelling results in both task incremental as well as cross-dataset continual learning across image and audio modalities, while being able to distinguish seen from unseen data. We have demonstrated that our approach readily benefits from recent model advances such as autoregressive models (Gulrajani et al., 2017; Chen et al., 2017) and therefore expect to apply our approach to more complicated data such as larger scale color images with further improvements in generative models. As our approach is extendible, we envision future work to encompass dynamical neural network expansion to increase representation capacity when task complexity increases (Yoon et al., 2018; Rusu et al., 2016; Zhou et al., 2012), combination with soft-targets (Hinton et al., 2014; Li & Hoiem, 2016) or transfer to entirely unsupervised scenarios where the classifier learns only dataset ids instead of distinct classes (Achille et al., 2018).

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

## A  CONTINUAL LEARNING 2-D LATENT SPACE VISUALIZATION

A natural consequence of our joint model with a shared probabilistic encoder is that the classifier encourages the formation of class-specific regions of high density in the latent space. During continual incremental learning, these regions keep shifting with every task increment while maintaining their class-specificity. New regions of high density emerge for newly added classes. As can be observed in figure 3, at the end of the first task two regions have been formed around the mean of the $\mathcal{N}(0,1)$ prior when training our CDVAE model on the MNIST (LeCun et al., 1998) dataset in a class-incremental upper-bound fashion. With every addition of the next classes, the latent embedding shifts around the mean of the prior to accommodate the new classes with distinct classes separated by regions of low density. Furthermore, it can also be seen in figures 3e and 3f that the shape and the location of the high density regions in the latent embedding are model dependent.

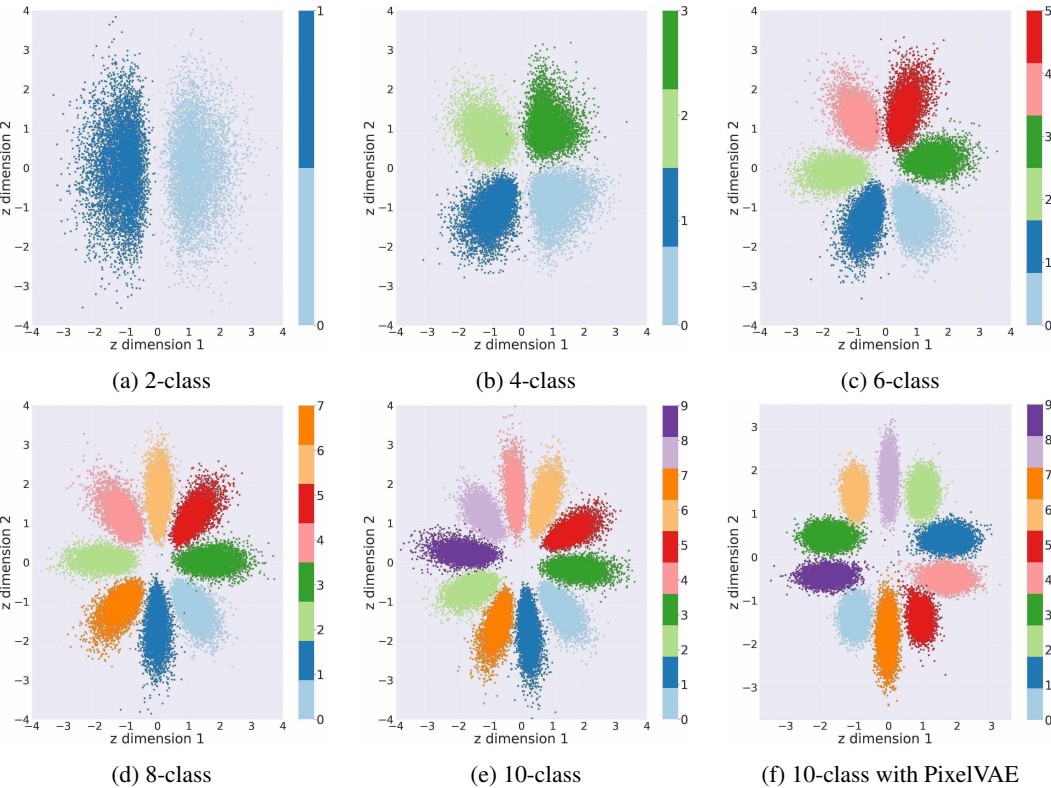

Figure 3: 2-D latent space visualization for continually learned incremental upper-bound MNIST at the end of every task increment for CDVAE (a-e) and at the end of training for all task increments for PixCDVAE (f).

# B    BALANCING CLASSIFICATION ACCURACY AND KULLBACK LEIBLER DIVERGENCE - THE ROLE OF $\beta$

In equations 1 and 2, we have added a weight term $\beta$ to the KL divergence to balance the individual loss terms, similar to the work of Zhou et al. (2012) and Higgins et al. (2017). In a standard $\beta$-VAE Higgins et al. (2017), $\beta$ can be seen as a hyper-parameter that balances the reconstruction accuracy with the distribution approximation quality in the latent space. As mentioned in the main body, in our work, this factor also regulates the trade-off between the additional constraint imposed by the classifier, needing to be able to linearly separate the classes given $z$, and the quality of the approximation to the training data. As reconstruction losses are typically dependent on the inputs' spatial dimensionality and the classification loss on the respective number of classes, we normalize the loss accordingly. This yields loss values whose scale is comparable and in practice changes the relative scale of the losses to each other. Whereas typically the reconstruction loss heavily outweighs the Kullback Leibler divergence and classification in non-normalized training for a $\beta$ value of $1.0$, this is not the case for our approach. Although the benefit is a more stable training irrespective of chosen dimensionalities, note that this however changes the role of specific $\beta$ values with respect to the original authors' Higgins et al. (2017) interpretation and the contribution of regularization.

To illustrate this we have trained our WRN model on MNIST with 2-D and 60-D latent spaces for multiple values of $\beta$ in tables 4 and 5 respectively. In these tables we report both the normalized

Table 4: Losses for the used WRN architecture with 2-D latent space and different $\beta$ values for MNIST. As stated in the main body, losses are normalized per dimension for training. Unnormalized values in nats are reported in brackets for reference purposes.

| 2-D latent | Beta | In nats per dimension (nats in brackets) | | | |
| | | KLD | Recon loss | Class Loss | Accuracy [%] |
|---|---|---|---|---|---|
| train | 1.0 | 1.039 (2.078) | 0.237 (185.8) | 0.539 (5.39) | 79.87 |
| test | | 1.030 (2.060) | 0.235 (184.3) | 0.596 (5.96) | 78.30 |
| train | 0.5 | 1.406 (2.812) | 0.230 (180.4) | 0.221 (2.21) | 93.88 |
| test | | 1.382 (2.764) | 0.228 (178.8) | 0.305 (3.05) | 92.07 |
| train | 0.1 | 2.055 (4.110) | 0.214 (167.8) | 0.042 (0.42) | 99.68 |
| test | | 2.071 (4.142) | 0.212 (166.3) | 0.116 (1.16) | 98.73 |
| train | 0.05 | 2.395 (4.790) | 0.208 (163.1) | 0.025 (0.25) | 99.83 |
| test | | 2.382 (4.764) | 0.206 (161.6) | 0.159 (1.59) | 98.79 |

Table 5: Losses for the used WRN architecture with 60-D latent space and different $\beta$ values for MNIST. As stated in the main body, losses are normalized per dimension for training. Unnormalized values in nats are reported in brackets for reference purposes.

| 60-D latent | Beta | In nats per dimension (nats in brackets) | | | |
| | | KLD | Recon loss | Class Loss | Accuracy [%] |
|---|---|---|---|---|---|
| train | 1.0 | 0.108 (6.480) | 0.184 (144.3) | 0.0110 (0.110) | 99.71 |
| test | | 0.110 (6.600) | 0.181 (142.0) | 0.0457 (0.457) | 99.03 |
| train | 0.5 | 0.151 (9.060) | 0.162 (127.1) | 0.0052 (0.052) | 99.87 |
| test | | 0.156 (9.360) | 0.159 (124.7) | 0.0451 (0.451) | 99.14 |
| train | 0.1 | 0.346 (20.76) | 0.124 (97.22) | 0.0022 (0.022) | 99.95 |
| test | | 0.342 (20.52) | 0.126 (98.79) | 0.0286 (0.286) | 99.38 |
| train | 0.05 | 0.476 (28.56) | 0.115 (90.16) | 0.0018 (0.018) | 99.95 |
| test | | 0.471 (28.26) | 0.118 (92.53) | 0.0311 (0.311) | 99.34 |

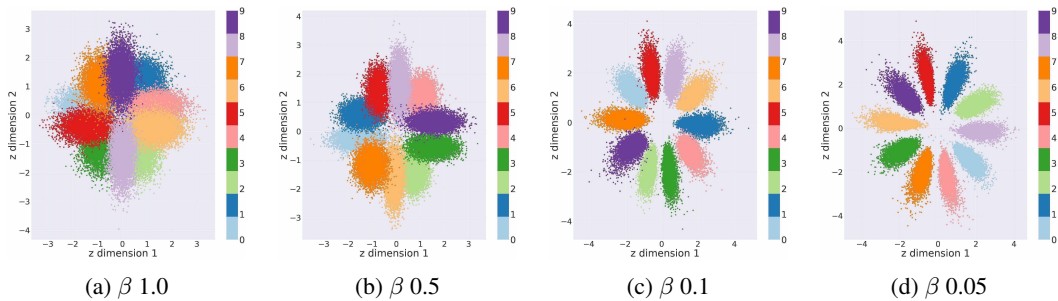

(a) $\beta$ 1.0          (b) $\beta$ 0.5          (c) $\beta$ 0.1          (d) $\beta$ 0.05

Figure 4: 2-D MNIST latent space visualization with different $\beta$ values for the used WRN architecture.

obtained losses during training, as well as the equivalent computed un-normalized losses as typically reported in the literature. While we have reported the latter for comparison purposes in our main body, the former are used in practice. We can observe that decreasing the value of beta leads to improvement of the classifier and reconstruction accuracy at the expense of KL divergence quality. In both 2-D and 60-D cases, a large beta value leads to significant underfit of the classifier. In order to enable adequate training of the classifier, a corresponding value of $\beta$ thus has to be chosen. Empirically we have found that a $\beta$ of $0.1$ works well universally. A nuance of the normalized loss is that $\beta$ values cannot be compared to ones in the un-normalized scenario. To pick a concrete example, the ratio between KLD and reconstruction loss with a $\beta$ value of $1.0$ for a 60-D latent space in table 5 is roughly $1 : 22$ without normalization, but approximately $1 : 1.6$ in our case. In practice our choice of $\beta$ would translate to much larger values in a conventional $\beta$-VAE.

However, the presence of the classifier, irrespective of the value of $\beta$ induces low density regions in latent space. For the 2-D latent space on a trained model for MNIST, we can observe that these low density regions become more apparent with smaller values of $\beta$, but manifest even for large values at the center due to the classifier's need to disentangle the class representations.

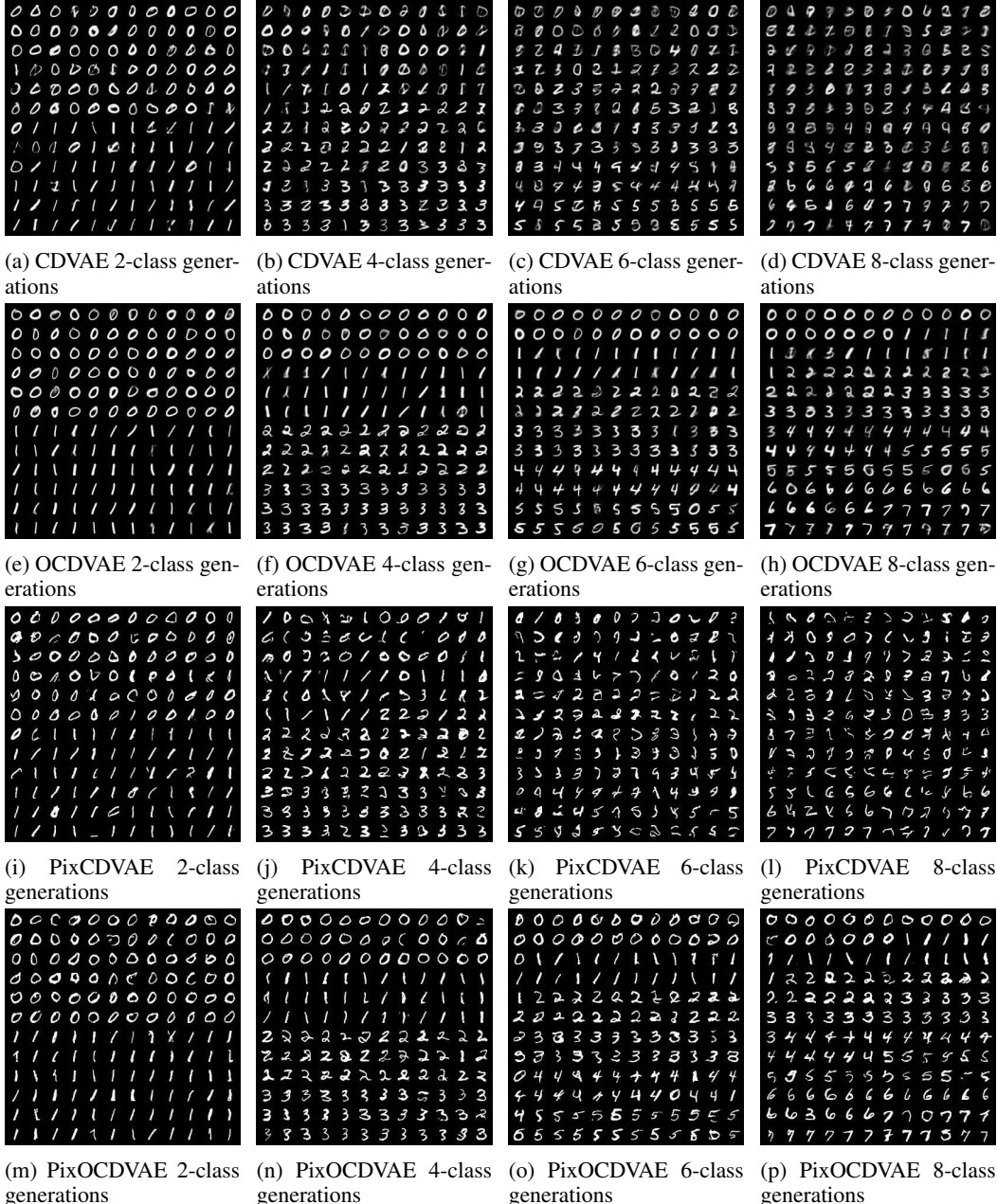

Figure 5: Generated images for continually learned incremental MNIST at the end of task increments for CDVAE (a-d), OCDVAE (e-h), PixCDVAE (i-l) and PixOCDVAE (m-p).

## C    GENERATIVE REPLAY EXAMPLES WITH CDVAE AND OCDVAE

As stated in section 2 of the main body, as well as exemplified in the previous section, the jointly optimized linear classifier directly affects the emergence of class specific areas of large probability density in the latent space. The effect of sampling from the prior without statistical outlier rejection for low density regions is shown in figure 5 for the MNIST dataset. For CDVAE/PixCDVAE we observe classifier confusion due to class interpolated examples, mentioned in section 2.2. As the generative model needs to learn how to replay old tasks' data based on its own former generations, this confusion and interpolations accumulate rapidly. This is not the case for OCDVAE/PixOCDVAE, where misclassifications are scarce and the generative model is capable of maintaining high visual

fidelity throughout continual training. As the OCDVAE constrains the sampling to regions of high density, in principle the generative replay could reproduce solely the original data without any interpolation akin to an over-fit. However, for the purpose of generative replay and estimating the log-likelihood of former seen data distributions, this can be desirable in the continual learning scenario as long as variety is ensured. Both our continual learning results presented in the main paper, as well as the visual examples of this section's figures indicate that this is the case. Similar tendencies can be observed for the other two datasets - FashionMNIST Xiao et al. (2017) (figure 6) and AudioMNIST (Becker et al., 2018) (figure 7). We note that we show AudioMNIST for the purpose of completeness as generated examples are difficult to interpret visually.

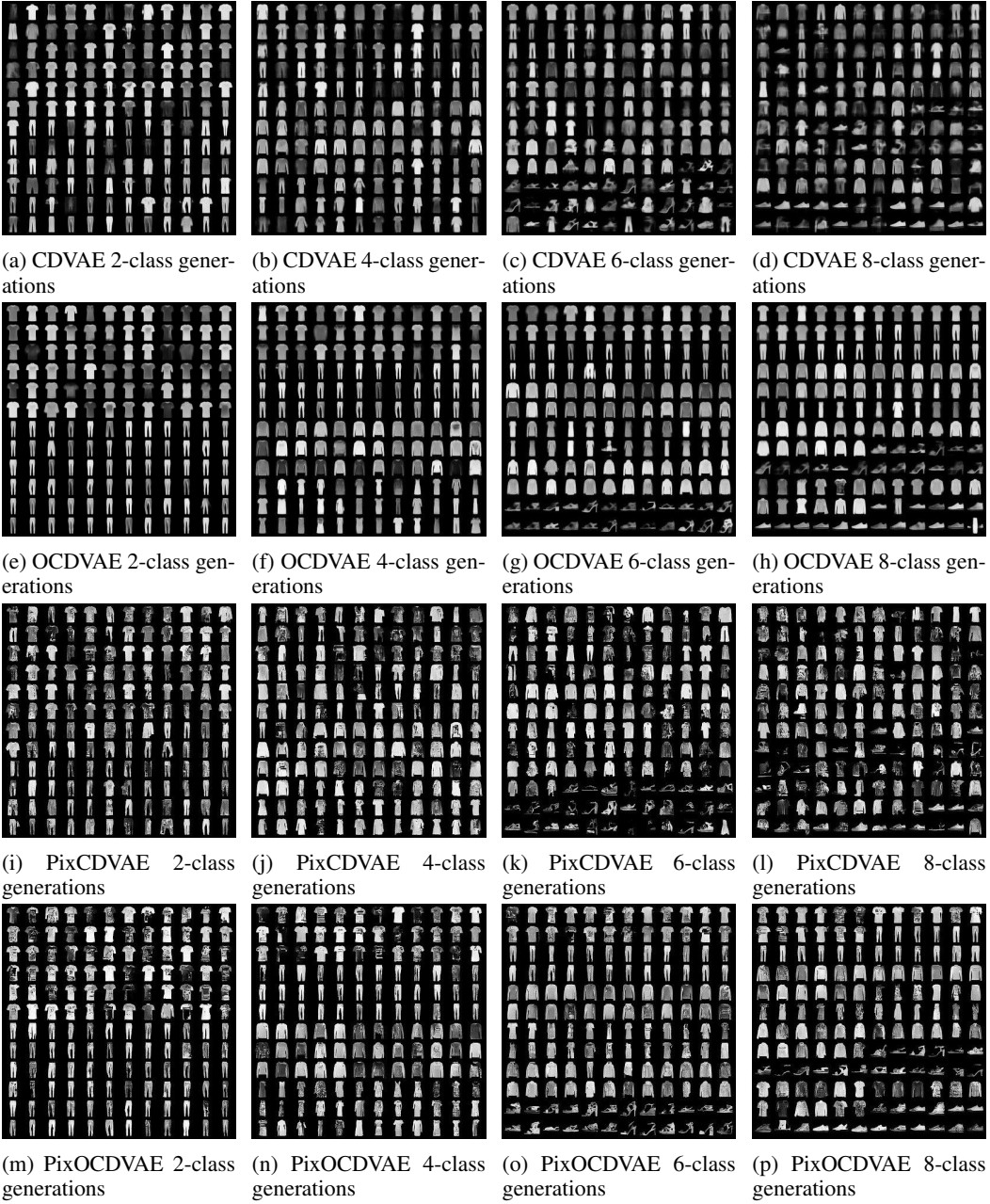

(a) CDVAE 2-class generations

(b) CDVAE 4-class generations

(c) CDVAE 6-class generations

(d) CDVAE 8-class generations

(e) OCDVAE 2-class generations

(f) OCDVAE 4-class generations

(g) OCDVAE 6-class generations

(h) OCDVAE 8-class generations

(i) PixCDVAE 2-class generations

(j) PixCDVAE 4-class generations

(k) PixCDVAE 6-class generations

(l) PixCDVAE 8-class generations

(m) PixOCDVAE 2-class generations

(n) PixOCDVAE 4-class generations

(o) PixOCDVAE 6-class generations

(p) PixOCDVAE 8-class generations

Figure 6: Generated images for continually learned incremental FashionMNIST at the end of task increments for CDVAE (a-d), OCDVAE (e-h), PixCDVAE (i-l) and PixOCDVAE (m-p).

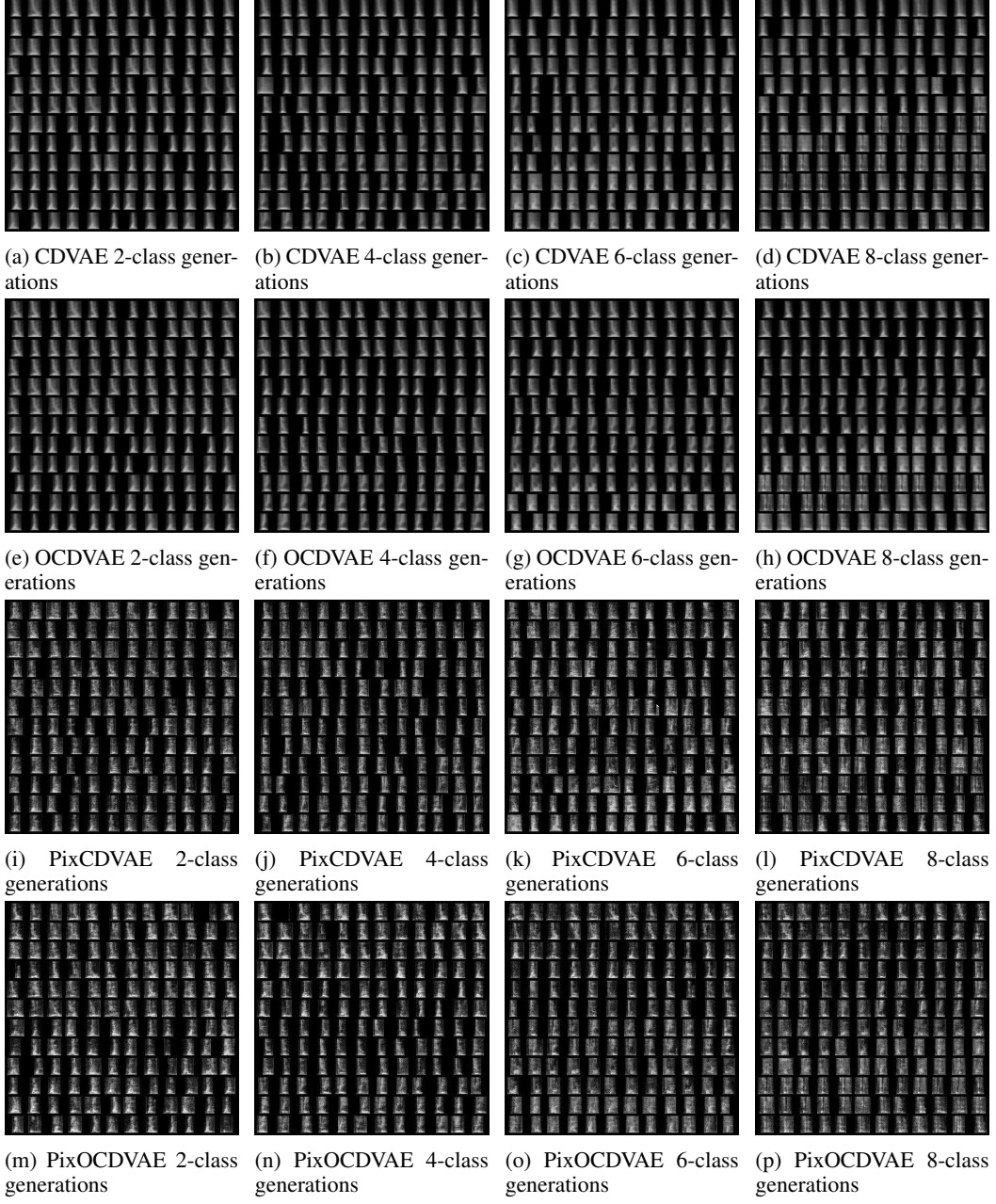

(a) CDVAE 2-class generations

(b) CDVAE 4-class generations

(c) CDVAE 6-class generations

(d) CDVAE 8-class generations

(e) OCDVAE 2-class generations

(f) OCDVAE 4-class generations

(g) OCDVAE 6-class generations

(h) OCDVAE 8-class generations

(i) PixCDVAE 2-class generations

(j) PixCDVAE 4-class generations

(k) PixCDVAE 6-class generations

(l) PixCDVAE 8-class generations

(m) PixOCDVAE 2-class generations

(n) PixOCDVAE 4-class generations

(o) PixOCDVAE 6-class generations

(p) PixOCDVAE 8-class generations

Figure 7: Generated images for continually learned incremental AudioMNIST at the end of task increments for CDVAE (a-d), OCDVAE (e-h), PixCDVAE (i-l) and PixOCDVAE (m-p).

# D  ILLUSTRATION OF GENERATIVE REPLAY WITH STATISTICAL OUTLIER REJECTION

In figure 8 we show generated images $x \sim p_{\phi,t}(\boldsymbol{x}|\boldsymbol{z})$ with $\boldsymbol{z} \sim p(\boldsymbol{z})$ and their corresponding class $c$ obtained from the classifier $p_{\boldsymbol{\xi},t}(y|\boldsymbol{z})$ for an OCDVAE model trained on the class incremental MNIST, after the last task increment $t = T$. Based on its sample from the prior, for each image we have further noted the open set statistical outlier probability $\omega_{c,t}$ from the respective class' Weibull model. Images are depicted in rows, whereas each row corresponds to a distinct class label. The figure exemplifies the sampling process for a mini-batch of size 128, where the amount of class samples within the mini-batch is not necessarily balanced. We observe how generated images that feature blurring and ambiguity are considered as strong statistical outliers, as well as examples with class interpolation and therefore hold a misclassified label. Using the latter examples to create a dataset for continual learning with generative replay hence entails accumulation of errors. In contrast to the conventional version with unconstrained sampling, our generative replay with statistical outlier rejection algorithm shown in algorithm 3 of the main body rejects these examples and prevents such errors to a large degree.

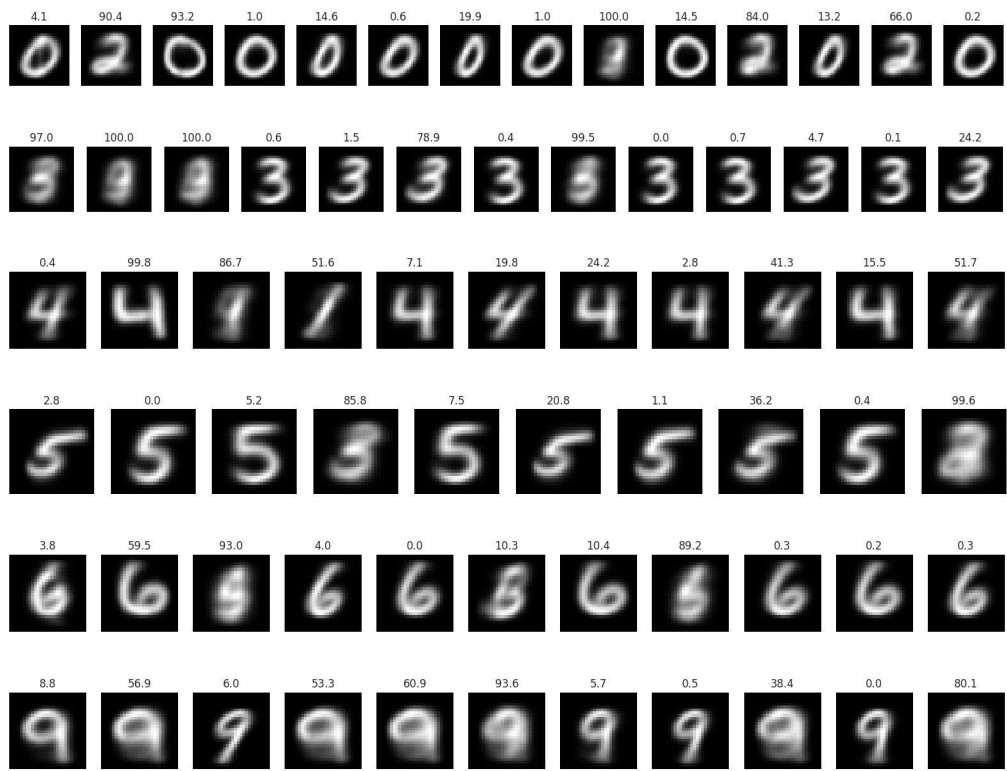

Figure 8: Illustration of generated images $x \sim p_{\phi,t}(\boldsymbol{x}|\boldsymbol{z})$ with $\boldsymbol{z} \sim p(\boldsymbol{z})$ and their corresponding class $c$ obtained from the classifier $p_{\boldsymbol{\xi},t}(y|\boldsymbol{z})$, together with their open set outlier probability $\omega_{c,t}$, for an OCDVAE model trained on incremental MNIST, after the last task increment $t = T$. From top to bottom the identified classes are: 0, 3, 4, 5, 6 and 9. It is observable how the statistical outlier probability is proportional to the degree of interpolation between classes, blur and thus ambiguity.

# E    FULL CLASS INCREMENTAL RESULTS

In addition to the comparative analysis provided in section 3.4 of the main body, we provide the class-incremental results for each of the three datasets at the end of every task increment, averaged over 5 experimental repetitions in tables 6, 7 and 8 respectively. These tables aid in making some additional observations about the behavior of the different continual learning algorithms across consecutive task increments.

We once again observe the increased effect of error accumulation due to unconstrained generative sampling from the prior in the CDVAE and the PixCDVAE models in comparison to their open set counterparts. The statistical deviations across experiment repetitions in the base and the overall classification accuracies are higher and are generally decreased by the open set models. For example, in table 6 the MNIST base and overall accuracy deviations of CDVAE are higher than the respective values for OCDVAE starting from the second task increment. Correspondingly, the accuracy values themselves experience larger decline for CDVAE than for OCDVAE with progressive increments. This difference is not as pronounced at the end of the first task increment because the models haven't been trained on any of their own generated data yet. Successful literature approaches such as the variational generative replay proposed by the authors of Farquhar & Gal (2018) thus avoid repeated learning based on previous generated examples and simply store and retain a separate generative model for each task. The strength of our model is that, instead of storing a trained model for each task increment, we are able to continually keep training our joint model with data generated for all previously seen tasks by filtering out ambiguous samples through statistical outlier rejection. Similar trends can also be observed for the respective pixel models.

## E.1    BACKWARD TRANSFER

The weight sharing and the presence of a generative expanding single-headed classifier open up the scope for both forward and backward transfer of knowledge in the continual learning context. Figure 9 shows an interesting case of the latter for class-incremental learning with our OCDVAE model on the AudioMNIST dataset. The addition of two new classes (four and five) at the end of the second increment leads to an improvement in the classification performance on class two, as indicated by the confusion matrices.

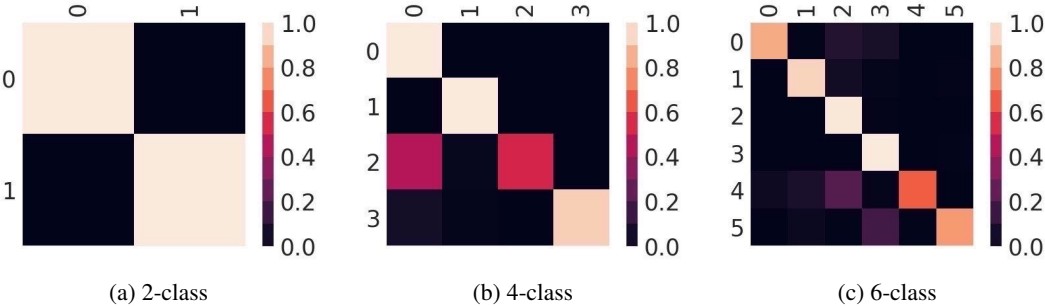

(a) 2-class                    (b) 4-class                    (c) 6-class

Figure 9: AudioMNIST confusion matrices for incrementally learned classes of the OCDVAE model. When adding classes two and three the model experiences difficulty in classification, however is able to overcome this challenge by exhibiting backward transfer when later learning classes four and five. It is also observable how forgetting of the initial classes is limited.

## E.2    PIXEL MODEL BITS PER DIMENSION CLASSIFICATION LOSSES

Although the main body reports PixelVAE reconstruction log-likelihoods in nats, these models are practically formulated as a classification problem with a 256-way Softmax. The corresponding loss is in bits per dimension. We have converted these values to have a better comparison, but in order to do so we need to sample from the pixel decoder's multinomial distribution to calculate a binary cross-entropy on reconstructed images. The bits per dimension classification loss values for our PixelVAE based experiments in the main body are provided for reference here. The PixCDVAE and PixOCDVAE achieve final losses on all tasks of $1.019 \pm 0.014$ and $1.047 \pm 0.010$ for MNIST,

Table 6: Results for class incremental continual learning approaches averaged over 5 runs, baselines and the reference isolated learning scenario for MNIST at the end of every task increment. $\alpha_t$ and $\gamma_t$ indicate the respective accuracy and NLL reconstruction metrics at the end of every task increment $t$. $KL_t$ denotes the corresponding KL divergence.

| MNIST | t | CDVAE ISO | CDVAE UB | CDVAE LB | EWC | Dual Model | Dual Pix Model | CDVAE | PixCDVAE | OCDVAE | PixOCDVAE |
|---|---|---|---|---|---|---|---|---|---|---|---|
| $\alpha_{base,t}$ (%) | 1 | | 100.0 | 100.0 | $99.88_{\pm 0.010}$ | $99.98_{\pm 0.023}$ | $99.97_{\pm 0.002}$ | $99.97_{\pm 0.029}$ | $99.97_{\pm 0.026}$ | $99.98_{\pm 0.018}$ | $99.86_{\pm 0.084}$ |
| | 2 | | 99.82 | 00.00 | $00.61_{\pm 0.057}$ | $99.77_{\pm 0.032}$ | $99.54_{\pm 0.285}$ | $97.28_{\pm 3.184}$ | $96.90_{\pm 2.907}$ | $99.30_{\pm 0.100}$ | $99.64_{\pm 0.095}$ |
| | 3 | | 99.80 | 00.00 | $00.17_{\pm 0.045}$ | $99.51_{\pm 0.094}$ | $99.16_{\pm 0.611}$ | $87.66_{\pm 8.765}$ | $90.12_{\pm 5.846}$ | $96.69_{\pm 2.173}$ | $98.88_{\pm 0.491}$ |
| | 4 | | 99.85 | 00.00 | $00.49_{\pm 0.017}$ | $98.90_{\pm 0.207}$ | $98.33_{\pm 1.119}$ | $54.70_{\pm 22.84}$ | $76.84_{\pm 9.095}$ | $94.71_{\pm 1.792}$ | $98.11_{\pm 0.797}$ |
| | 5 | | 99.57 | 00.00 | $00.45_{\pm 0.059}$ | $97.31_{\pm 0.489}$ | $98.04_{\pm 1.397}$ | $19.86_{\pm 7.396}$ | $56.53_{\pm 4.032}$ | $92.53_{\pm 4.485}$ | $97.44_{\pm 0.785}$ |
| $\alpha_{new,t}$ (%) | 1 | | 100.0 | 100.0 | $99.88_{\pm 0.010}$ | $99.98_{\pm 0.023}$ | $99.97_{\pm 0.002}$ | $99.97_{\pm 0.029}$ | $99.97_{\pm 0.026}$ | $99.98_{\pm 0.018}$ | $99.86_{\pm 0.084}$ |
| | 2 | | 99.80 | 99.85 | $99.70_{\pm 0.013}$ | $99.81_{\pm 0.062}$ | $99.71_{\pm 0.122}$ | $99.75_{\pm 0.127}$ | $99.74_{\pm 0.052}$ | $99.80_{\pm 0.126}$ | $99.82_{\pm 0.027}$ |
| | 3 | | 99.67 | 99.94 | $99.94_{\pm 0.002}$ | $99.48_{\pm 0.294}$ | $99.41_{\pm 0.084}$ | $99.63_{\pm 0.172}$ | $99.22_{\pm 0.082}$ | $99.61_{\pm 0.055}$ | $99.56_{\pm 0.092}$ |
| | 4 | | 99.49 | 100.0 | $99.87_{\pm 0.015}$ | $99.46_{\pm 0.315}$ | $98.61_{\pm 0.312}$ | $99.05_{\pm 0.470}$ | $97.84_{\pm 0.180}$ | $99.15_{\pm 0.032}$ | $98.80_{\pm 0.292}$ |
| | 5 | | 99.10 | 99.86 | $99.58_{\pm 0.052}$ | $98.59_{\pm 0.106}$ | $97.31_{\pm 0.575}$ | $99.00_{\pm 0.100}$ | $96.77_{\pm 0.337}$ | $99.06_{\pm 0.171}$ | $98.63_{\pm 0.430}$ |
| $\alpha_{all,t}$ (%) | 1 | | 100.0 | 100.0 | $99.88_{\pm 0.010}$ | $99.98_{\pm 0.023}$ | $99.97_{\pm 0.002}$ | $99.97_{\pm 0.029}$ | $99.97_{\pm 0.026}$ | $99.98_{\pm 0.018}$ | $99.86_{\pm 0.084}$ |
| | 2 | | 99.81 | 49.92 | $50.16_{\pm 0.029}$ | $99.79_{\pm 0.049}$ | $99.60_{\pm 0.142}$ | $98.54_{\pm 1.638}$ | $98.37_{\pm 1.448}$ | $99.55_{\pm 0.036}$ | $99.69_{\pm 0.051}$ |
| | 3 | | 99.72 | 31.35 | $33.42_{\pm 0.027}$ | $99.32_{\pm 0.057}$ | $98.93_{\pm 0.291}$ | $95.01_{\pm 3.162}$ | $96.14_{\pm 1.836}$ | $98.46_{\pm 0.903}$ | $99.20_{\pm 0.057}$ |
| | 4 | | 99.50 | 24.82 | $25.36_{\pm 0.025}$ | $98.56_{\pm 0.021}$ | $98.22_{\pm 0.560}$ | $81.50_{\pm 9.369}$ | $91.25_{\pm 0.992}$ | $97.06_{\pm 1.069}$ | $98.13_{\pm 0.281}$ |
| | 5 | 99.45 | 99.29 | 20.16 | $20.26_{\pm 0.027}$ | $96.64_{\pm 0.079}$ | $96.52_{\pm 0.658}$ | $64.34_{\pm 4.903}$ | $83.61_{\pm 0.927}$ | $93.24_{\pm 3.742}$ | $96.84_{\pm 0.346}$ |
| $\gamma_{base,t}$ (nats) or (bits/dim) | 1 | | 63.18 | 62.08 | | $62.17_{\pm 0.979}$ | $90.52_{\pm 0.263}$ | $64.34_{\pm 2.054}$ | $100.0_{\pm 1.572}$ | $62.53_{\pm 1.166}$ | $99.77_{\pm 2.768}$ |
| | 2 | | 62.85 | 126.8 | | $63.69_{\pm 0.576}$ | $91.27_{\pm 0.789}$ | $74.41_{\pm 10.89}$ | $100.4_{\pm 1.964}$ | $65.68_{\pm 1.166}$ | $101.2_{\pm 3.601}$ |
| | 3 | | 63.36 | 160.4 | | $67.34_{\pm 0.445}$ | $91.92_{\pm 0.991}$ | $81.89_{\pm 10.09}$ | $100.3_{\pm 4.562}$ | $69.29_{\pm 1.541}$ | $101.1_{\pm 4.014}$ |
| | 4 | | 64.25 | 126.9 | | $70.41_{\pm 0.436}$ | $91.75_{\pm 1.136}$ | $90.62_{\pm 10.08}$ | $102.7_{\pm 7.134}$ | $71.69_{\pm 1.379}$ | $101.0_{\pm 4.573}$ |
| | 5 | | 64.99 | 123.2 | | $75.08_{\pm 0.623}$ | $92.05_{\pm 1.212}$ | $101.6_{\pm 8.347}$ | $102.4_{\pm 6.195}$ | $77.16_{\pm 1.104}$ | $100.5_{\pm 4.942}$ |
| $\gamma_{new,t}$ (nats) or (bits/dim) | 1 | | 63.18 | 62.08 | | $62.17_{\pm 0.979}$ | $90.52_{\pm 0.263}$ | $64.34_{\pm 2.054}$ | $100.0_{\pm 1.572}$ | $62.53_{\pm 1.166}$ | $99.77_{\pm 2.768}$ |
| | 2 | | 88.75 | 87.93 | | $88.03_{\pm 0.664}$ | $115.8_{\pm 0.805}$ | $89.91_{\pm 0.107}$ | $125.7_{\pm 2.413}$ | $89.64_{\pm 3.709}$ | $124.6_{\pm 3.822}$ |
| | 3 | | 82.53 | 87.22 | | $83.46_{\pm 0.992}$ | $107.7_{\pm 0.600}$ | $87.65_{\pm 0.530}$ | $118.3_{\pm 3.523}$ | $85.37_{\pm 1.725}$ | $116.5_{\pm 2.219}$ |
| | 4 | | 72.68 | 74.61 | | $73.23_{\pm 0.280}$ | $100.9_{\pm 0.659}$ | $79.49_{\pm 0.489}$ | $107.1_{\pm 5.316}$ | $74.75_{\pm 0.777}$ | $102.3_{\pm 1.844}$ |
| | 5 | | 85.88 | 92.00 | | $89.32_{\pm 0.626}$ | $113.4_{\pm 0.820}$ | $93.55_{\pm 0.391}$ | $118.2_{\pm 1.572}$ | $89.68_{\pm 0.618}$ | $113.3_{\pm 0.755}$ |
| $\gamma_{all,t}$ (nats) or (bits/dim) | 1 | | 63.18 | 62.08 | | $62.17_{\pm 0.979}$ | $90.52_{\pm 0.263}$ | $64.34_{\pm 2.054}$ | $100.0_{\pm 1.572}$ | $62.53_{\pm 1.166}$ | $99.77_{\pm 2.768}$ |
| | 2 | | 75.97 | 107.3 | | $75.64_{\pm 0.600}$ | $102.9_{\pm 0.408}$ | $82.02_{\pm 5.488}$ | $111.9_{\pm 2.627}$ | $76.62_{\pm 1.695}$ | $112.7_{\pm 3.300}$ |
| | 3 | | 79.58 | 172.3 | | $81.24_{\pm 0.262}$ | $104.8_{\pm 1.114}$ | $89.88_{\pm 3.172}$ | $114.9_{\pm 4.590}$ | $82.95_{\pm 1.878}$ | $114.6_{\pm 4.788}$ |
| | 4 | | 79.72 | 203.1 | | $82.92_{\pm 0.489}$ | $103.9_{\pm 0.759}$ | $95.83_{\pm 2.747}$ | $114.3_{\pm 3.963}$ | $85.30_{\pm 1.524}$ | $112.1_{\pm 2.150}$ |
| | 5 | 78.12 | 81.97 | 163.7 | | $88.29_{\pm 0.363}$ | $106.1_{\pm 0.868}$ | $107.6_{\pm 1.724}$ | $118.7_{\pm 5.320}$ | $92.92_{\pm 2.283}$ | $111.9_{\pm 2.663}$ |
| $KL_{all,t}$ (nats) | 1 | | 12.55 | 13.08 | | $11.81_{\pm 0.123}$ | $1.410_{\pm 0.181}$ | $13.00_{\pm 0.897}$ | $5.629_{\pm 3.749}$ | $13.68_{\pm 0.785}$ | $5.635_{\pm 3.739}$ |
| | 2 | | 18.50 | 25.84 | | $16.15_{\pm 0.149}$ | $3.177_{\pm 0.702}$ | $20.20_{\pm 1.188}$ | $9.238_{\pm 0.674}$ | $18.01_{\pm 0.154}$ | $7.495_{\pm 0.738}$ |
| | 3 | | 20.16 | 24.28 | | $16.46_{\pm 0.122}$ | $4.923_{\pm 1.085}$ | $24.24_{\pm 1.974}$ | $12.13_{\pm 0.977}$ | $20.02_{\pm 0.161}$ | $10.17_{\pm 1.528}$ |
| | 4 | | 20.48 | 26.32 | | $16.09_{\pm 0.177}$ | $5.603_{\pm 1.250}$ | $27.01_{\pm 1.851}$ | $14.32_{\pm 1.040}$ | $20.26_{\pm 0.186}$ | $11.66_{\pm 1.004}$ |
| | 5 | 22.12 | 21.02 | 24.87 | | $16.13_{\pm 0.225}$ | $9.296_{\pm 1.346}$ | $30.61_{\pm 1.240}$ | $16.37_{\pm 0.970}$ | $21.02_{\pm 0.717}$ | $12.49_{\pm 0.551}$ |

Table 7: Results for class incremental continual learning approaches averaged over 5 runs, baselines and the reference isolated learning scenario for FashionMNIST at the end of every task increment. $\alpha_t$ and $\gamma_t$ indicate the respective accuracy and NLL reconstruction metrics at the end of every task increment $t$. $KL_t$ denotes the corresponding KL divergence.

| Fashion | t | CDVAE ISO | CDVAE UB | CDVAE LB | EWC | Dual Model | Dual Pix Model | CDVAE | PixCDVAE | OCDVAE | PixOCDVAE |
|---|---|---|---|---|---|---|---|---|---|---|---|
| $\alpha_{base,t}$ (%) | 1 | | 99.65 | 99.60 | $99.17_{\pm 0.037}$ | $99.58_{\pm 0.062}$ | $99.57_{\pm 0.091}$ | $99.55_{\pm 0.035}$ | $99.58_{\pm 0.076}$ | $99.59_{\pm 0.082}$ | $99.54_{\pm 0.079}$ |
| | 2 | | 96.70 | 00.00 | $02.40_{\pm 0.122}$ | $94.50_{\pm 0.389}$ | $82.40_{\pm 6.688}$ | $92.02_{\pm 1.175}$ | $90.06_{\pm 1.782}$ | $92.36_{\pm 2.092}$ | $88.60_{\pm 1.998}$ |
| | 3 | | 95.95 | 00.00 | $01.63_{\pm 0.032}$ | $94.88_{\pm 0.432}$ | $78.55_{\pm 3.964}$ | $79.26_{\pm 4.170}$ | $83.70_{\pm 3.571}$ | $83.90_{\pm 2.310}$ | $87.66_{\pm 0.375}$ |
| | 4 | | 91.35 | 00.00 | $00.33_{\pm 0.097}$ | $82.25_{\pm 4.782}$ | $54.69_{\pm 3.853}$ | $50.16_{\pm 6.658}$ | $50.23_{\pm 7.004}$ | $64.70_{\pm 2.580}$ | $68.31_{\pm 3.308}$ |
| | 5 | | 92.20 | 00.00 | $00.17_{\pm 0.076}$ | $94.26_{\pm 0.192}$ | $60.04_{\pm 5.151}$ | $39.51_{\pm 7.173}$ | $47.83_{\pm 13.41}$ | $60.63_{\pm 12.16}$ | $74.45_{\pm 2.889}$ |
| $\alpha_{new,t}$ (%) | 1 | | 99.65 | 99.60 | $99.17_{\pm 0.037}$ | $99.58_{\pm 0.062}$ | $99.57_{\pm 0.091}$ | $99.55_{\pm 0.035}$ | $99.58_{\pm 0.076}$ | $99.59_{\pm 0.082}$ | $99.54_{\pm 0.079}$ |
| | 2 | | 95.55 | 97.95 | $96.09_{\pm 0.260}$ | $89.31_{\pm 0.311}$ | $97.73_{\pm 1.113}$ | $90.98_{\pm 0.626}$ | $96.47_{\pm 0.596}$ | $92.64_{\pm 2.302}$ | $97.31_{\pm 0.475}$ |
| | 3 | | 93.35 | 99.95 | $99.92_{\pm 0.012}$ | $86.06_{\pm 2.801}$ | $99.09_{\pm 0.367}$ | $90.26_{\pm 1.435}$ | $97.33_{\pm 0.725}$ | $83.40_{\pm 3.089}$ | $96.88_{\pm 1.156}$ |
| | 4 | | 84.75 | 99.90 | $99.95_{\pm 0.060}$ | $73.63_{\pm 3.861}$ | $97.55_{\pm 0.588}$ | $85.65_{\pm 2.127}$ | $96.12_{\pm 0.675}$ | $84.18_{\pm 2.715}$ | $95.47_{\pm 1.332}$ |
| | 5 | | 97.50 | 99.80 | $99.60_{\pm 0.023}$ | $93.55_{\pm 0.708}$ | $98.85_{\pm 0.141}$ | $96.92_{\pm 0.774}$ | $97.91_{\pm 0.596}$ | $96.51_{\pm 0.707}$ | $98.63_{\pm 0.176}$ |
| $\alpha_{all,t}$ (%) | 1 | | 99.65 | 99.60 | $99.17_{\pm 0.037}$ | $99.58_{\pm 0.062}$ | $99.57_{\pm 0.091}$ | $99.55_{\pm 0.035}$ | $99.58_{\pm 0.076}$ | $99.59_{\pm 0.082}$ | $99.54_{\pm 0.079}$ |
| | 2 | | 95.75 | 48.97 | $49.28_{\pm 0.242}$ | $91.91_{\pm 0.043}$ | $86.22_{\pm 3.704}$ | $91.83_{\pm 0.730}$ | $92.93_{\pm 0.160}$ | $92.31_{\pm 1.163}$ | $92.17_{\pm 1.425}$ |
| | 3 | | 93.02 | 33.33 | $34.34_{\pm 0.009}$ | $79.98_{\pm 0.634}$ | $76.77_{\pm 4.378}$ | $83.35_{\pm 1.597}$ | $84.07_{\pm 1.069}$ | $86.93_{\pm 0.870}$ | $87.30_{\pm 0.322}$ |
| | 4 | | 87.51 | 25.00 | $25.21_{\pm 0.100}$ | $64.37_{\pm 0.707}$ | $62.93_{\pm 3.738}$ | $64.66_{\pm 3.204}$ | $64.42_{\pm 1.837}$ | $76.05_{\pm 1.391}$ | $76.36_{\pm 1.267}$ |
| | 5 | 89.54 | 89.24 | 19.97 | $20.06_{\pm 0.059}$ | $63.21_{\pm 1.957}$ | $72.41_{\pm 2.941}$ | $58.82_{\pm 2.521}$ | $63.05_{\pm 1.826}$ | $69.88_{\pm 1.712}$ | $80.85_{\pm 0.721}$ |
| $\gamma_{base,t}$ (nats) or (bits/dim) | 1 | | 209.7 | 209.8 | | $207.7_{\pm 1.558}$ | $267.8_{\pm 1.246}$ | $208.9_{\pm 1.213}$ | $230.8_{\pm 3.024}$ | $209.7_{\pm 3.655}$ | $232.0_{\pm 2.159}$ |
| | 2 | | 207.4 | 240.7 | | $209.0_{\pm 0.731}$ | $273.6_{\pm 0.631}$ | $212.7_{\pm 0.579}$ | $232.5_{\pm 1.582}$ | $212.1_{\pm 0.937}$ | $231.8_{\pm 0.416}$ |
| | 3 | | 207.6 | 258.7 | | $213.0_{\pm 1.854}$ | $274.0_{\pm 0.552}$ | $219.5_{\pm 1.376}$ | $235.6_{\pm 2.784}$ | $216.9_{\pm 1.208}$ | $231.6_{\pm 0.832}$ |
| | 4 | | 207.7 | 243.6 | | $213.6_{\pm 0.509}$ | $273.7_{\pm 0.504}$ | $223.8_{\pm 0.837}$ | $236.4_{\pm 3.157}$ | $217.1_{\pm 0.979}$ | $231.4_{\pm 2.550}$ |
| | 5 | | 208.4 | 306.5 | | $217.7_{\pm 1.510}$ | $274.1_{\pm 0.349}$ | $232.8_{\pm 5.048}$ | $241.1_{\pm 1.747}$ | $222.8_{\pm 1.632}$ | $234.1_{\pm 1.498}$ |
| $\gamma_{new,t}$ (nats) or (bits/dim) | 1 | | 209.7 | 209.8 | | $207.7_{\pm 1.558}$ | $267.8_{\pm 1.246}$ | $208.9_{\pm 1.213}$ | $230.8_{\pm 3.024}$ | $209.7_{\pm 3.655}$ | $232.0_{\pm 2.159}$ |
| | 2 | | 241.1 | 240.2 | | $238.7_{\pm 0.081}$ | $313.4_{\pm 1.006}$ | $241.8_{\pm 0.902}$ | $275.8_{\pm 1.888}$ | $241.9_{\pm 0.960}$ | $275.3_{\pm 1.473}$ |
| | 3 | | 213.6 | 211.8 | | $211.6_{\pm 0.543}$ | $269.1_{\pm 0.616}$ | $215.4_{\pm 0.501}$ | $268.3_{\pm 3.852}$ | $213.0_{\pm 0.635}$ | $262.9_{\pm 1.893}$ |
| | 4 | | 220.5 | 219.7 | | $219.5_{\pm 0.216}$ | $282.4_{\pm 0.321}$ | $223.6_{\pm 0.381}$ | $259.1_{\pm 1.305}$ | $220.9_{\pm 0.522}$ | $259.6_{\pm 2.050}$ |
| | 5 | | 246.2 | 242.0 | | $242.8_{\pm 0.898}$ | $305.8_{\pm 0.286}$ | $248.8_{\pm 0.398}$ | $283.2_{\pm 2.150}$ | $244.0_{\pm 0.646}$ | $283.5_{\pm 2.458}$ |
| $\gamma_{all,t}$ (nats) or (bits/dim) | 1 | | 209.7 | 209.8 | | $207.7_{\pm 1.558}$ | $267.8_{\pm 1.246}$ | $208.9_{\pm 1.213}$ | $230.8_{\pm 3.024}$ | $209.7_{\pm 3.655}$ | $232.0_{\pm 2.159}$ |
| | 2 | | 224.2 | 240.4 | | $223.8_{\pm 0.402}$ | $293.8_{\pm 0.349}$ | $226.6_{\pm 2.31}$ | $254.3_{\pm 1.513}$ | $226.9_{\pm 0.918}$ | $255.8_{\pm 0.436}$ |
| | 3 | | 220.7 | 246.1 | | $221.9_{\pm 0.648}$ | $285.7_{\pm 0.510}$ | $227.2_{\pm 0.606}$ | $261.5_{\pm 2.970}$ | $224.9_{\pm 0.642}$ | $259.1_{\pm 0.929}$ |
| | 4 | | 220.4 | 238.7 | | $225.1_{\pm 3.629}$ | $284.9_{\pm 0.703}$ | $230.4_{\pm 0.524}$ | $263.2_{\pm 2.259}$ | $226.1_{\pm 0.560}$ | $259.5_{\pm 3.218}$ |
| | 5 | 224.8 | 226.2 | 275.1 | | $230.5_{\pm 1.543}$ | $289.5_{\pm 0.396}$ | $242.2_{\pm 0.754}$ | $271.7_{\pm 2.117}$ | $234.6_{\pm 0.823}$ | $267.2_{\pm 0.586}$ |
| $KL_{all,t}$ (nats) | 1 | | 12.17 | 12.20 | | $9.710_{\pm 0.345}$ | $3.610_{\pm 0.856}$ | $13.21_{\pm 0.635}$ | $7.164_{\pm 0.759}$ | $13.28_{\pm 0.644}$ | $7.809_{\pm 1.255}$ |
| | 2 | | 16.54 | 17.47 | | $10.65_{\pm 0.101}$ | $6.247_{\pm 0.710}$ | $17.60_{\pm 0.755}$ | $13.79_{\pm 0.282}$ | $15.56_{\pm 0.696}$ | $12.23_{\pm 0.287}$ |
| | 3 | | 18.84 | 19.34 | | $11.34_{\pm 0.057}$ | $7.811_{\pm 0.799}$ | $21.25_{\pm 0.872}$ | $18.26_{\pm 0.818}$ | $17.35_{\pm 0.307}$ | $15.36_{\pm 0.530}$ |
| | 4 | | 20.06 | 17.31 | | $10.96_{\pm 0.106}$ | $8.982_{\pm 0.812}$ | $25.21_{\pm 0.929}$ | $21.75_{\pm 0.561}$ | $19.81_{\pm 0.462}$ | $18.31_{\pm 0.333}$ |
| | 5 | 23.27 | 20.27 | 21.61 | | $11.45_{\pm 0.228}$ | $9.781_{\pm 1.068}$ | $26.68_{\pm 0.859}$ | $22.14_{\pm 0.377}$ | $20.47_{\pm 0.742}$ | $17.93_{\pm 0.360}$ |

Table 8: Results for class incremental continual learning approaches averaged over 5 runs, baselines and the reference isolated learning scenario for AudioMNIST at the end of every task increment. $\alpha_t$ and $\gamma_t$ indicate the respective accuracy and NLL reconstruction metrics at the end of every task increment $t$. $KL_t$ denotes the corresponding KL divergence.

| Audio | t | CDVAE ISO | CDVAE UB | CDVAE LB | EWC | Dual Model | Dual Pix Model | CDVAE | PixCDVAE | OCDVAE | PixOCDVAE |
|---|---|---|---|---|---|---|---|---|---|---|---|
| $\alpha_{base,t}$ (%) | 1 | | 99.99 | 100.0 | $100.0_{\pm 0.000}$ | $100.0_{\pm 0.000}$ | $100.0_{\pm 0.000}$ | $99.21_{\pm 0.568}$ | $99.71_{\pm 0.218}$ | $99.95_{\pm 0.035}$ | $99.27_{\pm 0.410}$ |
| | 2 | | 99.92 | 00.00 | $00.16_{\pm 0.040}$ | $93.08_{\pm 5.854}$ | $99.52_{\pm 0.273}$ | $98.98_{\pm 0.766}$ | $97.86_{\pm 0.799}$ | $98.61_{\pm 0.490}$ | $97.88_{\pm 2.478}$ |
| | 3 | | 100.0 | 00.00 | $00.29_{\pm 0.029}$ | $83.25_{\pm 6.844}$ | $93.15_{\pm 3.062}$ | $92.44_{\pm 1.306}$ | $81.38_{\pm 5.433}$ | $95.12_{\pm 2.248}$ | $95.82_{\pm 3.602}$ |
| | 4 | | 99.92 | 00.00 | $00.31_{\pm 0.015}$ | $72.02_{\pm 0.677}$ | $81.55_{\pm 8.468}$ | $76.43_{\pm 4.715}$ | $50.58_{\pm 14.60}$ | $86.37_{\pm 5.63}$ | $91.56_{\pm 5.640}$ |
| | 5 | | 98.42 | 00.00 | $00.11_{\pm 0.007}$ | $61.57_{\pm 0.747}$ | $64.60_{\pm 8.739}$ | $59.36_{\pm 7.147}$ | $29.94_{\pm 18.47}$ | $79.73_{\pm 4.070}$ | $75.25_{\pm 10.18}$ |
| $\alpha_{new,t}$ (%) | 1 | | 99.99 | 100.0 | $100.0_{\pm 0.000}$ | $100.0_{\pm 0.000}$ | $100.0_{\pm 0.000}$ | $99.21_{\pm 0.568}$ | $99.71_{\pm 0.218}$ | $99.95_{\pm 0.035}$ | $99.27_{\pm 0.410}$ |
| | 2 | | 99.75 | 100.0 | $99.78_{\pm 0.019}$ | $86.25_{\pm 8.956}$ | $99.71_{\pm 0.043}$ | $91.82_{\pm 4.577}$ | $99.78_{\pm 0.128}$ | $89.23_{\pm 7.384}$ | $99.81_{\pm 0.189}$ |
| | 3 | | 98.92 | 99.58 | $99.25_{\pm 0.054}$ | $95.16_{\pm 1.490}$ | $98.23_{\pm 1.092}$ | $95.20_{\pm 1.495}$ | $98.41_{\pm 0.507}$ | $94.43_{\pm 3.030}$ | $99.30_{\pm 0.550}$ |
| | 4 | | 97.33 | 98.67 | $97.03_{\pm 0.019}$ | $62.52_{\pm 4.022}$ | $95.31_{\pm 0.868}$ | $53.02_{\pm 6.132}$ | $94.30_{\pm 0.914}$ | $72.22_{\pm 8.493}$ | $97.87_{\pm 0.293}$ |
| | 5 | | 98.67 | 100.0 | $99.41_{\pm 0.207}$ | $89.41_{\pm 0.691}$ | $98.18_{\pm 0.885}$ | $84.93_{\pm 6.297}$ | $97.00_{\pm 0.520}$ | $89.52_{\pm 6.586}$ | $99.43_{\pm 0.495}$ |
| $\alpha_{all,t}$ (%) | 1 | | 99.99 | 100.0 | $100.0_{\pm 0.000}$ | $100.0_{\pm 0.000}$ | $100.0_{\pm 0.000}$ | $99.21_{\pm 0.568}$ | $99.71_{\pm 0.218}$ | $99.95_{\pm 0.035}$ | $99.27_{\pm 0.410}$ |
| | 2 | | 99.83 | 50.00 | $50.16_{\pm 0.119}$ | $89.67_{\pm 1.763}$ | $99.50_{\pm 0.157}$ | $93.84_{\pm 2.558}$ | $98.64_{\pm 0.875}$ | $93.93_{\pm 3.756}$ | $99.67_{\pm 0.033}$ |
| | 3 | | 99.56 | 33.19 | $33.28_{\pm 0.022}$ | $78.24_{\pm 3.315}$ | $95.37_{\pm 1.750}$ | $94.26_{\pm 1.669}$ | $90.10_{\pm 1.431}$ | $95.70_{\pm 1.524}$ | $97.77_{\pm 1.017}$ |
| | 4 | | 98.60 | 24.58 | $24.50_{\pm 4.209}$ | $60.43_{\pm 4.209}$ | $86.97_{\pm 2.797}$ | $77.90_{\pm 4.210}$ | $75.55_{\pm 3.891}$ | $85.59_{\pm 3.930}$ | $95.41_{\pm 1.345}$ |
| | 5 | 97.75 | 97.87 | 20.02 | $19.98_{\pm 0.032}$ | $47.42_{\pm 1.447}$ | $75.50_{\pm 3.032}$ | $81.49_{\pm 1.944}$ | $63.44_{\pm 5.252}$ | $87.72_{\pm 1.594}$ | $90.23_{\pm 1.139}$ |
| $\gamma_{base,t}$ (nats) or (bits/dim) | 1 | | 433.7 | 423.2 | | $422.3_{\pm 0.573}$ | $434.2_{\pm 1.068}$ | $435.2_{\pm 15.69}$ | $432.6_{\pm 0.321}$ | $424.2_{\pm 2.511}$ | $433.8_{\pm 0.370}$ |
| | 2 | | 422.5 | 439.4 | | $426.6_{\pm 2.840}$ | $434.4_{\pm 1.082}$ | $423.9_{\pm 0.517}$ | $432.5_{\pm 0.551}$ | $425.2_{\pm 1.402}$ | $433.5_{\pm 1.464}$ |
| | 3 | | 420.7 | 429.2 | | $425.0_{\pm 0.339}$ | $434.6_{\pm 0.785}$ | $422.7_{\pm 0.690}$ | $432.9_{\pm 0.723}$ | $423.8_{\pm 1.148}$ | $433.1_{\pm 1.269}$ |
| | 4 | | 419.9 | 428.5 | | $425.4_{\pm 0.081}$ | $434.2_{\pm 1.209}$ | $422.8_{\pm 0.367}$ | $433.0_{\pm 0.781}$ | $423.5_{\pm 0.937}$ | $433.0_{\pm 1.283}$ |
| | 5 | | 418.4 | 432.9 | | $425.2_{\pm 0.244}$ | $435.1_{\pm 1.915}$ | $422.7_{\pm 0.182}$ | $431.4_{\pm 0.666}$ | $423.5_{\pm 0.586}$ | $432.3_{\pm 0.189}$ |
| $\gamma_{new,t}$ (nats) or (bits/dim) | 1 | | 433.7 | 423.2 | | $422.3_{\pm 0.573}$ | $434.2_{\pm 1.068}$ | $435.2_{\pm 15.69}$ | $432.6_{\pm 0.321}$ | $424.2_{\pm 2.511}$ | $433.8_{\pm 0.370}$ |
| | 2 | | 381.2 | 384.1 | | $381.3_{\pm 2.039}$ | $390.4_{\pm 0.694}$ | $382.5_{\pm 1.355}$ | $389.4_{\pm 0.208}$ | $385.3_{\pm 12.56}$ | $389.4_{\pm 1.304}$ |
| | 3 | | 435.9 | 436.7 | | $436.8_{\pm 0.188}$ | $444.7_{\pm 0.545}$ | $436.3_{\pm 0.639}$ | $442.7_{\pm 0.513}$ | $436.9_{\pm 0.688}$ | $442.4_{\pm 0.275}$ |
| | 4 | | 485.9 | 487.1 | | $486.5_{\pm 0.432}$ | $497.4_{\pm 0.740}$ | $486.7_{\pm 0.385}$ | $494.4_{\pm 0.700}$ | $486.5_{\pm 0.701}$ | $494.8_{\pm 0.386}$ |
| | 5 | | 421.3 | 425.2 | | $422.4_{\pm 0.784}$ | $431.9_{\pm 1.032}$ | $423.9_{\pm 0.681}$ | $428.0_{\pm 0.851}$ | $422.9_{\pm 0.537}$ | $429.7_{\pm 1.223}$ |
| $\gamma_{all,t}$ (nats) or (bits/dim) | 1 | | 433.7 | 423.2 | | $422.3_{\pm 0.573}$ | $434.2_{\pm 1.068}$ | $435.2_{\pm 15.69}$ | $432.6_{\pm 0.321}$ | $424.2_{\pm 2.511}$ | $433.8_{\pm 0.370}$ |
| | 2 | | 401.9 | 411.8 | | $404.0_{\pm 2.407}$ | $412.4_{\pm 0.871}$ | $403.2_{\pm 0.831}$ | $410.9_{\pm 0.351}$ | $403.5_{\pm 1.274}$ | $411.5_{\pm 1.406}$ |
| | 3 | | 412.1 | 418.9 | | $414.4_{\pm 0.385}$ | $423.3_{\pm 0.618}$ | $413.6_{\pm 0.410}$ | $421.0_{\pm 1.026}$ | $413.8_{\pm 0.573}$ | $421.9_{\pm 0.661}$ |
| | 4 | | 430.3 | 438.4 | | $433.9_{\pm 0.374}$ | $441.6_{\pm 0.420}$ | $432.4_{\pm 0.436}$ | $439.8_{\pm 0.833}$ | $432.6_{\pm 0.862}$ | $439.8_{\pm 0.718}$ |
| | 5 | 429.7 | 427.2 | 440.4 | | $432.7_{\pm 0.385}$ | $440.3_{\pm 1.297}$ | $431.4_{\pm 0.255}$ | $436.9_{\pm 0.751}$ | $430.9_{\pm 0.541}$ | $437.7_{\pm 0.432}$ |
| $KL_{all,t}$ (nats) | 1 | | 11.65 | 11.20 | | $4.639_{\pm 0.107}$ | $4.361_{\pm 0.671}$ | $11.78_{\pm 1.478}$ | $9.293_{\pm 0.943}$ | $11.16_{\pm 0.713}$ | $11.87_{\pm 1.504}$ |
| | 2 | | 11.78 | 13.61 | | $5.135_{\pm 0.127}$ | $5.130_{\pm 0.636}$ | $15.13_{\pm 1.128}$ | $14.00_{\pm 0.748}$ | $14.06_{\pm 1.140}$ | $12.40_{\pm 0.719}$ |
| | 3 | | 13.40 | 17.09 | | $5.427_{\pm 0.105}$ | $5.399_{\pm 0.724}$ | $18.18_{\pm 1.140}$ | $20.28_{\pm 0.774}$ | $13.61_{\pm 0.901}$ | $14.41_{\pm 0.461}$ |
| | 4 | | 13.61 | 14.41 | | $5.243_{\pm 0.135}$ | $5.817_{\pm 1.038}$ | $22.93_{\pm 1.134}$ | $24.91_{\pm 0.845}$ | $17.58_{\pm 1.102}$ | $16.00_{\pm 0.505}$ |
| | 5 | 17.89 | 15.15 | 14.52 | | $5.470_{\pm 0.055}$ | $6.031_{\pm 0.832}$ | $22.96_{\pm 0.912}$ | $27.14_{\pm 1.139}$ | $18.52_{\pm 1.131}$ | $17.45_{\pm 0.835}$ |

$2.851 \pm 0.0026$ and $2.852 \pm 0.0047$ for FashionMNIST, $4.425 \pm 0.0010$ and $4.451 \pm 0.0198$ for AudioMNIST. For cross-dataset experiments starting with FashionMNIST first, the corresponding loss values in bits per dimension for PixCDVAE are $2.260 \pm 0.0078$ and $2.238 \pm 0.0021$ for PixOCD-VAE. In the reverse direction the values are $2.232 \pm 0.0177$ and $2.218 \pm 0.0014$ respectively. In the dual model scenario, the separate PixVAE achieves final losses on all tasks of $1.040 \pm 0.0103$ for MNIST, $2.242 \pm 0.0124$ for FashionMNIST and $4.406 \pm 0.0024$ for AudioMNIST. Loss values for the cross dataset setting are $2.253 \pm 0.0047$ when FashionMNIST gets trained first and $2.279 \pm 0.0104$ for the reverse direction starting with AudioMNIST.

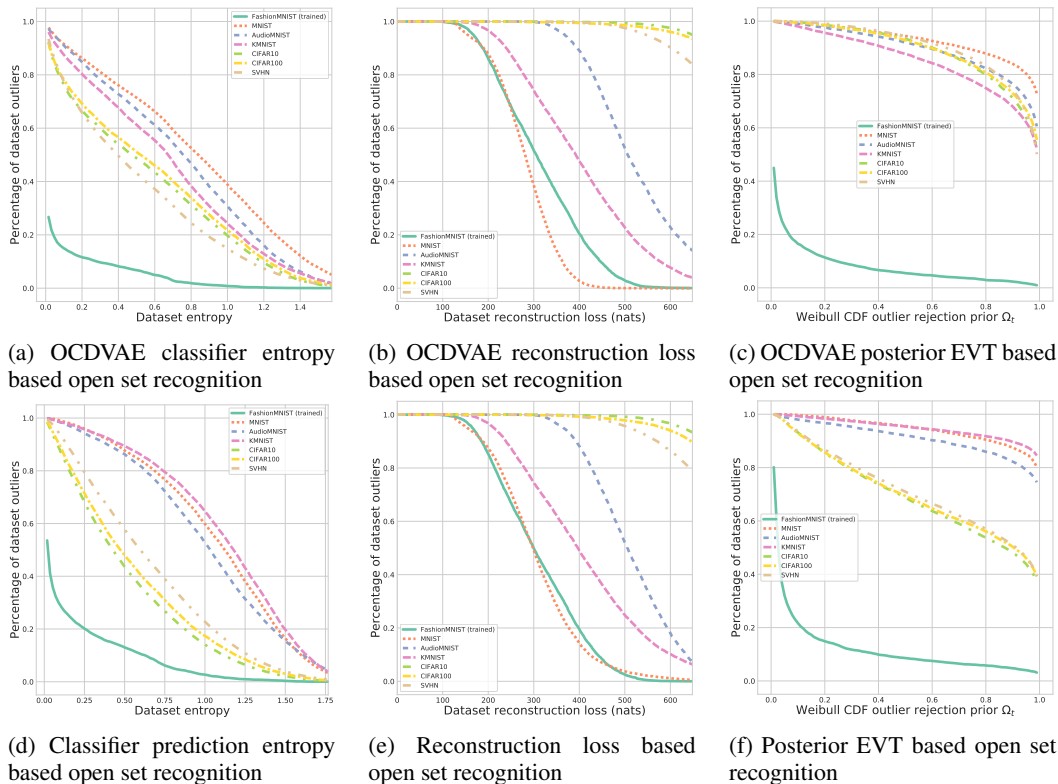

(a) OCDVAE classifier entropy based open set recognition

(b) OCDVAE reconstruction loss based open set recognition

(c) OCDVAE posterior EVT based open set recognition

(d) Classifier prediction entropy based open set recognition

(e) Reconstruction loss based open set recognition

(f) Posterior EVT based open set recognition

Figure 10: Trained FashionMNIST OCDVAE evaluated on unseen datasets. All metrics are reported as the mean over 100 approximate posterior samples per data point. (a) The classifier entropy values by itself are insufficient to separate most of unknown from the known task's test data. (b) Reconstruction loss allows for a partial distinction. (c) Our posterior-based open set recognition considers the large majority of unknown data as statistical outliers across a wide range of rejection priors $\Omega_t$. (d-f) Corresponding outlier detection using the two separate models of the dual-model approach.

# F  ADDITIONAL OPEN SET RECOGNITION VISUALIZATION

As we point out in section 3 of the main paper, our posterior based open set recognition considers almost all of the unknown datasets as statistical outliers, while at the same time regarding unseen test data from the originally trained tasks as distribution inliers across a wide range of rejection priors. In addition to the outlier rejection curves for FashionMNIST and the quantitative results presented in the main body, we also show the full outlier rejection curves for the remaining datasets, as well as all dual model approaches in figures 10, 11 and 12. These figures visually support the quantitative findings described in the main body and respective conclusions. In summary, the joint OCDVAE performs better at open set recognition in direct comparison to the dual model setting, particularly when using the EVT based criterion. Apart from the MNIST dataset, where reconstruction loss can be a sufficient metric for open set detection, the latent based approach also exhibits less dependency on the outlier rejection prior and consistently improves the ability to discern unknown data.

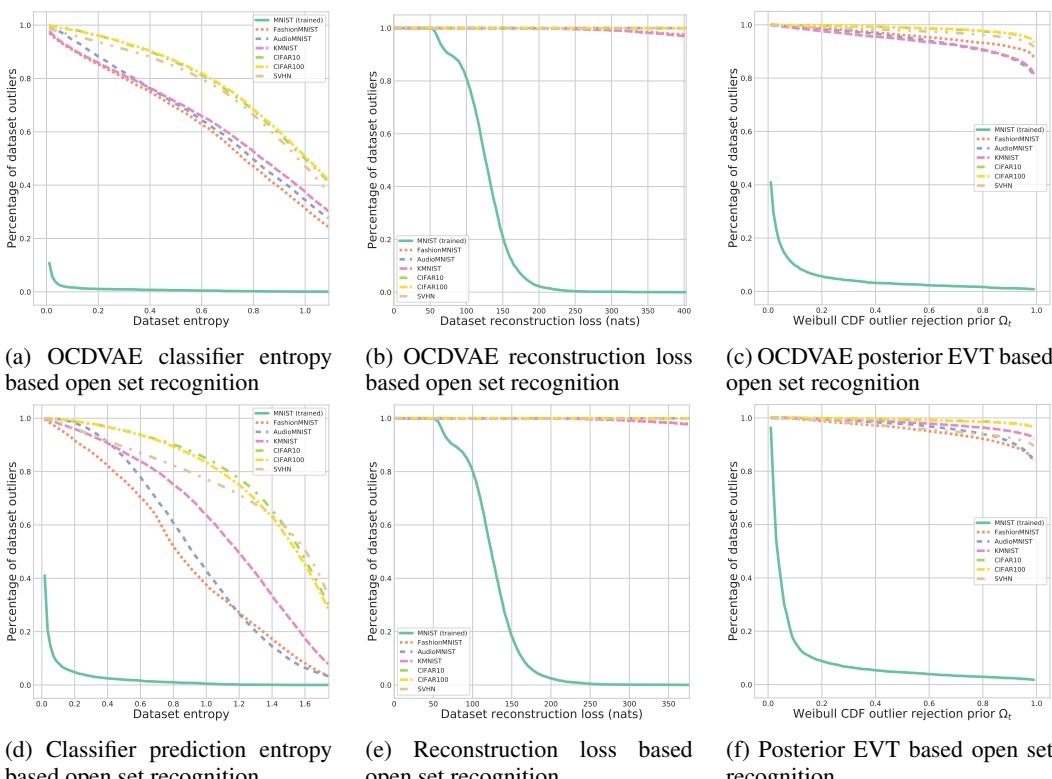

(a) OCDVAE classifier entropy based open set recognition

(b) OCDVAE reconstruction loss based open set recognition

(c) OCDVAE posterior EVT based open set recognition

(d) Classifier prediction entropy based open set recognition

(e) Reconstruction loss based open set recognition

(f) Posterior EVT based open set recognition

Figure 11: Trained MNIST OCDVAE evaluated on unseen datasets. All metrics are reported as the mean over $100$ approximate posterior samples per data point. (a) The classifier entropy values by itself are insufficient to separate most of unknown from the known task's test data. (b) Reconstruction loss allows for distinction if the cut-off is chosen correctly. (c) Our posterior-based open set recognition considers the large majority of unknown data as statistical outliers across a wide range of rejection priors $\Omega_t$. (d-f) Corresponding outlier detection using the two separate models of the dual-model approach.

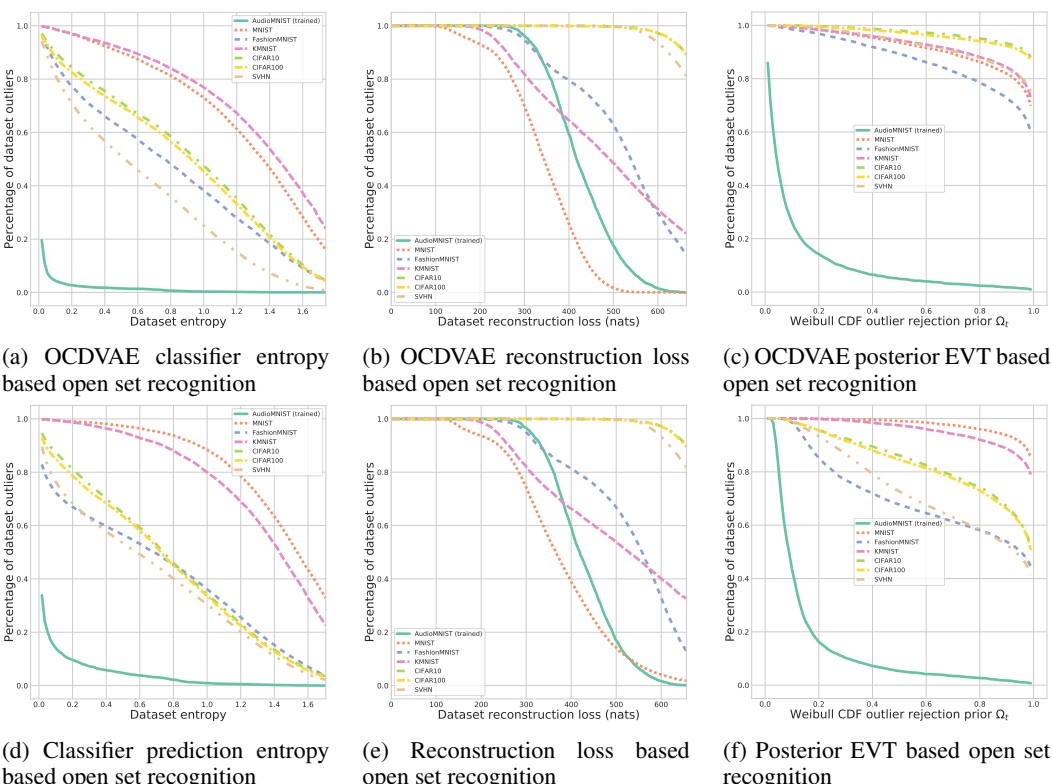

(a) OCDVAE classifier entropy based open set recognition

(b) OCDVAE reconstruction loss based open set recognition

(c) OCDVAE posterior EVT based open set recognition

(d) Classifier prediction entropy based open set recognition

(e) Reconstruction loss based open set recognition

(f) Posterior EVT based open set recognition

Figure 12: Trained AudioMNIST OCDVAE evaluated on unseen datasets. All metrics are reported as the mean over 100 approximate posterior samples per data point. (a) The classifier entropy values by itself are insufficient to separate most of unknown from the known task's test data. (b) Reconstruction loss allows for a partial distinction. (c) Our posterior-based open set recognition considers the large majority of unknown data as statistical outliers across a wide range of rejection priors $\Omega_t$. (d-f) Corresponding outlier detection using the two separate models of the dual-model approach.

## G ARCHITECTURE DEFINITIONS AND ADDITIONAL HYPER-PARAMETERS

Our previous description of the training hyperparameters in the main text is extended here by specifying the exact encoder and decoder architecture, and the additional hyperparameters for the Adam (Kingma & Ba, 2015) optimizer used for training in each of our evaluated methods. We also provide the hyperparameter values necessary for evaluating EWC in the class-incremental learning and cross-dataset scenarios.

We point the reader to tables 9 and 10 for detailed encoder and decoder configurations. For the autoregressive addition to our joint model, we set the number of output channels of the decoder to 60 and append 3 pixel decoder layers, each with a kernel size of $7 \times 7$ and 60 channels. The hyperparameters for Adam optimization include a $\beta_1$ of 0.9, $\beta_2$ of 0.999 and $\epsilon$ of $10^{-8}$.

For the EWC experiments, the number of Fisher samples is fixed to the total number of data points from all the previously seen tasks. A suitable Fisher multiplier ($\lambda$) value has been determined by conducting a grid search over a set of five values: 50, 100, 500, 1000 and 5000 on held-out validation data for the first two tasks in sequence. We observe exploding gradients if $\lambda$ is too high. However, a very small $\lambda$ leads to excessive drift in the network weight distribution across subsequent tasks that further results in catastrophic inference. A balance between these two phenomena is achieved for a $\lambda$ value of 500 in the class-incremental scenario and 1000 in the cross-dataset setting.

Table 9: 14-layer WRN encoder with a widen factor of 10. Convolutional layers (conv) are parametrized by a quadratic filter size followed by the amount of filters. p and s represent padding and stride respectively. If no padding or stride is specified then p = 0 and s = 1. Skip connections are an additional operation at a layer, with the layer to be skipped specified in brackets. Every convolutional layer is followed by batch-normalization and a ReLU.

| Layer type | WRN encoder | |
| --- | --- | --- |
| Layer 1 | conv $3 \times 3$ - 48, p = 1 | |
| Block 1 | conv $3 \times 3$ - 160, p = 1;
conv $3 \times 3$ - 160, p = 1
conv $3 \times 3$ - 160, p = 1;
conv $3 \times 3$ - 160, p = 1 | conv $1 \times 1$ - 160 (skip next layer)

shortcut (skip next layer) |
| Block 2 | conv $3 \times 3$ - 320, s = 2, p = 1;
conv $3 \times 3$ - 320, p = 1
conv $3 \times 3$ - 320, p = 1;
conv $3 \times 3$ - 320, p = 1 | conv $1 \times 1$ - 320, s = 2 (skip next layer)

shortcut (skip next layer) |
| Block 3 | conv $3 \times 3$ - 640, s = 2, p = 1;
conv $3 \times 3$ - 640, p = 1
conv $3 \times 3$ - 640, p = 1;
conv $3 \times 3$ - 640, p = 1 | conv $1 \times 1$ - 640, s = 2 (skip next layer)

shortcut (skip next layer) |

Table 10: 15-layered WRN decoder with a widen factor of 10. P refers to the quadratic input's spatial dimension. Convolutional (conv) and transposed convolutional (conv_t) layers are parametrized by a quadratic filter size followed by the amount of filters. p and s represent padding and stride respectively. If no padding or stride is specified then p = 0 and s = 1. Skip connections are an additional operation at a layer, with the layer to be skipped specified in brackets. Every convolutional and fully-connected (FC) layer are followed by batch-normalization and a ReLU. The model ends on a linear transformation with a Sigmoid function.

| Layer type | WRN decoder | |
|---|---|---|
| Layer 1 | FC $640 \times \lfloor * \rfloor P/4 \times \lfloor * \rfloor P/4$ | |
| Block 1 | conv_t $3 \times 3$ - 320, p = 1; 
 conv $3 \times 3$ - 320, p = 1 | conv_t $1 \times 1$ - 320 (skip next layer) |
| | conv $3 \times 3$ - 320, p = 1; 
 conv $3 \times 3$ - 320, p = 1 
 upsample $\times 2$ | shortcut (skip next layer) |
| Block 2 | conv_t $3 \times 3$ - 160, p = 1; 
 conv $3 \times 3$ - 160, p = 1 | conv_t $1 \times 1$ - 160 (skip next layer) |
| | conv $3 \times 3$ - 160, p = 1; 
 conv $3 \times 3$ - 160, p = 1 
 upsample $\times 2$ | shortcut (skip next layer) |
| Block 3 | conv_t $3 \times 3$ - 48, p = 1; 
 conv $3 \times 3$ - 48, p = 1 | conv_t $1 \times 1$ - 48 (skip next layer) |
| | conv $3 \times 3$ - 48, p = 1; 
 conv $3 \times 3$ - 48, p = 1 | shortcut (skip next layer) |
| Layer 2 | conv $3 \times 3$ - 3, p = 1 | |

