# OpenReview forum: "Unified Probabilistic Deep Continual Learning through Generative Replay and Open Set Recognition"
_ICLR.cc/2020/Conference — Reject_

### Official Review · AnonReviewer1 · 2019-10-23
**Official Blind Review #1**

**Rating:** 3

**Review:**

This paper tackles the problem of catastrophic forgetting when data is organized in a large number of batches of data (tasks) that are sequentially made available. To avoid catastrophic forgetting, the authors learn a VAE that generates the training data (both inputs and labels) and retrain it using samples from the new task combined with samples generated from the VAE trained in the previous tasks (generative replay). In this way, there's no need to store all past data and even the first learned batch keeps being refreshed and should not be forgotten.

I like that this paper uses a single global probabilistic model instead of separate discriminative and generative ones. Unfortunately, there are several things that left me unconvinced about this paper:

1) Presentation of the paper

- Variables x, y, z are introduced and talked about without explanation. The graphical model or factorization assumptions are not even mentioned until after the loss has been defined. A normal flow is to first describe the model and what the involved variables mean, and then talk about what the loss for learning it should be, not the other way around.

- Text contradicting the equation: "In order to balance the individual loss terms, we normalize according to dimensions and weight the KL divergence with a constant of 0.1". But equation (2) shows a loss with no weighting. I'm assuming the text is correct, but then a beta should be added to the equation in front of the KL divergence.

- Tables and figures are inconveniently far from where they are referenced in the text.

2) Theoretical inconsistencies

Although the system might work overall, two things seem to be technically incorrect:

- The decoder and classifier are expected to approximate the distribution of training data according to the authors (for valid generative replay). This is not true in a beta-VAE. The weighting of the KL that the authors introduce is going to bias the learned generator towards the high probability regions. This is not a sound mechanism to achieve an as-faithful-as-possible (limited by the expressiveness of the encoder-decoder architectures) approximation to the training data.

- A Weibull distribution is used to model the same data, again, in a different way. I.e., there are two different probabilistic models modeling the same data in inconsistent ways and one or the other is used depending on the part of the system. (As an example, q(z) could be arbitrarily multimodal as far as the encoder is concerned, but the Weibull seems to force one mode per class. But regardless of this, both models are inconsistent.)

- Similarly, the proposed rejection sampling scheme of OCDVAE is not consistent with the theory of VAEs and it's a post-hoc tweak that is not theoretically expected to provide a pdf of data with lower KL divergence to the true data pdf.

3) Experiments

Finally, the experimental results do not look very compelling, it seems to be overall worse than the baselines in the two image datasets and slightly better in the audio dataset, so it's unclear that this approach is superior.

**Experience Assessment:**

I have published one or two papers in this area.

**Review Assessment: Checking Correctness Of Derivations And Theory:**

I assessed the sensibility of the derivations and theory.

**Review Assessment: Checking Correctness Of Experiments:**

I assessed the sensibility of the experiments.

**Review Assessment: Thoroughness In Paper Reading:**

I read the paper at least twice and used my best judgement in assessing the paper.

---

> ### Author Response · Authors · 2019-11-08
> **Reviewer 1 comments**
>
> Dear reviewer, thank you for taking the time to assess our paper. Please first read our preamble before proceeding with the specific comments.
>
>
> Re-emphasizing our contribution:
>
> We unfortunately feel like your review is solely focused on the aspect of generative replay. While the rejection replay is one part of our contribution, the open set mechanism of our single probabilistic model primarily allows for natural out-of-distribution (OOD) detection (algorithm 2).
> In other words, our model can prevent nonsensical predictions for unknown classes. We quantitatively demonstrate OOD detection in section 3.4 and table 3, where our suggested approach fares much better than separate discriminative and generative models and OOD detection based on conventional metrics such as predictive entropy or reconstruction loss (even with uncertainty).
> This is a unique aspect of our paper that the current continual learning literature doesn’t include and we believe it is very important for robust systems.
>
>
> 1) Presentation of the paper
>
> “Flow of the model section”
> We agree with your assessment. We have used up the strict page limit of 10 pages and thus have concentrated on first introducing generative replay, as we thought the majority of readers to be familiar with the VAE framework. We believe the flow of this section can easily be changed in the revised version.
>
> “But equation (2) shows a loss with no weighting”
> You are right in the assumption that the text is correct. We sincerely apologize for this and are correcting the equations in the revision.
>
> “Tables and figures are inconveniently far from where they are referenced“
> We will place the tables and figures closer to their point of reference in the revised version.
>
>
> 2) Theoretical inconsistencies
>
> "KL weighting and proposed rejection sampling scheme"
>
> In our model there is the additional constraint of the linear classifier that needs to be able to linearly separate the classes (given z) in order to perform well. This introduction of the classification loss is in direct competition with the KL divergence term. If the weight value gives too little preference to the classification loss, it is going to suffer in accuracy. If the classification loss is heavily favored, the quality of the approximation to the training data is less important. High density regions are going to be formed independently of whether KL is weighted with a beta <= 1 or other way around the classifier weighted with an alpha >= 1. Our proposed rejection sampling scheme thus makes sure to avoid usage of samples from low density regions. To improve understanding, we have decided to add an appendix section, where we demonstrate the trade-off between KLD and classification loss and provide complementary 2-D latent space visualizations.
>
> “A Weibull distribution is used to model the same data, again, in a different way. I.e., there are two different probabilistic models modeling the same data in inconsistent ways”
>
> The Weibull distribution does not actively influence the training of the generative model. Our proposed approach is a meta-recognition method that uses knowledge of the probability distribution of extreme values of sample distances from the mode. Due to the variational approximation, the estimated pdf will have non-zero probabilities for latent vector values that do not represent samples that have been seen before. For predictions on an unseen unknown data instance, conventional metrics fail to disregard such samples and avoid subsequent overconfident nonsensical classifier predictions. This is where the heavy tail distance distribution can provide bounds in order to gauge a classifier prediction’s trustworthiness. This is necessary and independent of the quality of the approximation to the training data.
>
>
> 3) “the experimental results do not look very compelling, it seems to be overall worse than the baselines...”
>
> Unfortunately we do not understand the origin of this statement. For the primary purpose of maintaining classification accuracies during continual learning, our proposed approach performs at least on par or outperforms other methods. Naturally the values for upper-bound and isolated experiments, where real data is present, are better. The KLD for the dual model approach with a separate VAE achieves a better approximation as it is not concerned about the classification task at hand. However, the aim of this work is to solve the classification task in a continual learning setting in a robust fashion with respect to unseen unknown instances. In addition to this, the accuracy of OOD detection achieved by our model significantly outperforms all other approaches, including separate models and different metrics such as predictive entropy or reconstruction loss (even with uncertainty). We kindly request you to revisit the results, specifically also the OOD detection results of table 3, and provide us with further explanation of why you think of our overall results as not compelling.

---

### Official Review · AnonReviewer2 · 2019-10-24
**Official Blind Review #2**

**Rating:** 3

**Review:**

This paper combines replay and openMax approach to help continual learning.  The results shows robustness on different dataset include image and audio in the continual learning condition, where the new come data has a different distribution but the model still able to maintain reasonable quality for the previously and newly come examples. To my understanding, this approach was not ground-breaking but seems a reasonable combinations.

I'm learning to give weak reject for this paper because of it's poorly written. (1) It's very hard to align the contribution claimed by the paper and previous work in the introduction section. I highly suggest the author re-write this part and has a separate section about related work and explicit describe the difference compare to others. (2) The contribution seems over-claimed, it said it's a unified framework, but I don't understand what it has been unified. (3) In the experimental part, it use audioMNIST. Any reason to use this dataset? There are more well-defined audio task such as TIMIT for phoneme classification or Aurora for digital recognition. They are more easy to understand since they have well established benchmark.

Given my limited knowledge on this the literature of this topic, I'm happy to change the score if the written being improved and the following question being addressed.

(1)  Introduction. I take most of space to describe previous work, but hard to find out what's difference of this paper. My understanding it combines A and B and apply it to C. But it claim it's a unified framework.
(2) "We fully share our model across tasks and automatically expand the linear classifier with additional units when encountering new classes, thus not requiring explicit task labels." I cannot link "automatic" with the proposed method. Is that doable because of the proposed framework?
(3) Why use AudioMNIST which is an unusual task for audio?
(4) For the giant Table 1, I suggest link each acronym with the reference paper. So it can easily get how it associate with different approach. Highlight some numbers can also help the reader understand what's going on in this giant table.
(5) Can the author give me some insights, what these KL loss demonstrated in the table? I feel since you use the beta-vae version, the kl scale is depend on different approach, not really comparable for these different models.

**Experience Assessment:**

I do not know much about this area.

**Review Assessment: Checking Correctness Of Derivations And Theory:**

I assessed the sensibility of the derivations and theory.

**Review Assessment: Checking Correctness Of Experiments:**

I assessed the sensibility of the experiments.

**Review Assessment: Thoroughness In Paper Reading:**

I read the paper at least twice and used my best judgement in assessing the paper.

---

> ### Author Response · Authors · 2019-11-08
> **Reviewer 2 general comments (part 1 of our answer)**
>
> Thank you again for taking the time to review and comment our work. Please kindly read our preamble comments on general contributions before proceeding with the answers to your raised specific points.
> We have split our answer to your review in two parts. The first part addresses your remarks in the main text about our work’s novelty and unified framework. The second part answers your specific technical questions.
>
> Contribution and novelty: “This paper combines replay and openMax approach to help continual learning“
>
> We unfortunately believe that there is a misunderstanding with respect to our contributions and used methods. Specifically, we do not make use of the OpenMax approach that explicitly modifies Softmax outputs and rejects according to thresholds on the final modified Softmax confidence scores. We do not operate on the output space. Instead we draw inspiration from the derived extreme value theory (EVT) and adapt it to a Bayesian inference point of view by using it in the latent space of a generative model. This mechanism is further naturally used to improve samples drawn from the prior that are selected for generative replay in continual learning. In that sense it is not only to help continual learning but to have a system that is capable of detecting out-of-distribution (OOD) examples during inference while being able to learn continually in a single probabilistic model. We hope that the clarification of the essence of our contribution illustrates that our work is significantly different from pure unconstrained generative replay or the OpenMax approach.
>
> “The results shows robustness on different dataset include image and audio in the continual learning condition, where the new come data has a different distribution but the model still able to maintain reasonable quality for the previously and newly come examples. To my understanding, this approach was not ground-breaking but seems a reasonable combinations.”
>
> In addition to the above statement we would like to re-emphasize that our contribution is two-fold: we propose a natural out-of-distribution detection mechanism and at the same time use this mechanism to also improve generative replay in continual learning in a single probabilistic model. This is more than simply “maintaining the quality” of the model through improved generative replay, but in addition explicitly incorporating a mechanism to correctly identify OOD examples as “unseen unknowns” and thus adding the model capability to prevent an otherwise garbage output (e.g. since the class instance being computed might not be trained in the current stage of the classifier). In addition to the tables 1 and 2, showing the improvements in continual learning, we also demonstrate this quantitatively by correctly detecting entire datasets as OOD in table 3. There we have shown how our proposed approach is superior to conventional metrics such as predictive entropy or reconstruction loss, even when (epistemic) uncertainty is accounted for by taking 100 samples from the approximate posterior. We also show that our approach for OOD detection in our joint generative model outperforms a separate discriminative and generative model in OOD detection (even if an EVT approach is used there).
>
>
> Unified framework (question 1): “The contribution seems over-claimed, it said it's a unified framework, but I don't understand what it has been unified.”
>
> Following the arguments of the previous paragraphs: we believe that the outline of our contributions at the end of the introduction section explicitly mentions that our approach unifies a generative replay based continual learning approach with open set recognition in a single probabilistic model. To the very best of our knowledge, our paper is the first approach to present such a system. Other continual learning works focus solely on avoiding catastrophic forgetting and if one feeds an instance from an unknown dataset into these models for prediction, the model is going to predict a wrong label with high confidence (by definition because the classifier doesn’t know the class). In contrast, our model will flag the instance as out-of-distribution to avoid such prediction, as demonstrated in section 3.4 and table 3 of our paper. We thus do not believe our contribution to be over-claimed at all.
>
> Related work: “It's very hard to align the contribution claimed by the paper and previous work in the introduction section. I highly suggest the author re-write this part and has a separate section about related work and explicit describe the difference compare to others”
>
> We have tried to be as comprehensive as possible in the review of the literature and explicitly stated our contributions in bullet-points, given the strict 10-page limit. We will improve on the statements with respect to our contributions and provide a clearer distinction to existing literature in the revised version. This should make our contribution/novelty clearer to the reader.

---

> ### Author Response · Authors · 2019-11-08
> **Reviewer 2 specific comments (part 2 of our answer)**
>
> Question 2: "We fully share our model across tasks and automatically expand the linear classifier ... I cannot link "automatic" with the proposed method.”
>
> Our approach uses a single model with a single classifier. Whenever a new class is encountered this classifier grows by adding a new unit for prediction (automatically because a new label is encountered). This is referred to in the literature as a “single-head” approach. In contrast, most of the literature uses a “multi-head” approach, where for new concepts encountered in continual learning a completely separate classifier is added. While this might make the accuracy look much better because each classifier has a much easier task to solve by itself, it is unclear which classifier to use during inference as a “task label” is needed to infer the correct classifier. To give a concrete example: you might first train AudioMNIST and then FashionMNIST. When you now want to predict you will need to know which classifier to use as otherwise the model might predict that an audio sample corresponds to e.g. a T-shirt.
> In our work, when feeding a new input to the classifier, we do not need any label to make a prediction as we only have a single classifier that contains all the tasks. If the input belongs to a known task we simply directly obtain the prediction. At the same time, if the input is statistically unlike to what the model has seen before, our proposed OOD detection mechanism will flag it as an unseen unknown class and a garbage prediction can be avoided. Tying back to the aspect of novelty, we are unaware of any other continual learning model that can achieve this.
>
> Question 3: “In the experimental part, it use audioMNIST. Any reason to use this dataset?”
>
> We agree that there might be other datasets and applications that researchers might be interested in. We have chosen the specific dataset because we were able to down-sample the spectrograms to 32x32 size without losing accuracy with respect to the original author’s experiments. This in return allows us to then use a single model across the visual and audio datasets in continual learning. This way we can highlight that our approach is capable of cross-domain and multi-modal continual learning and out-of-distribution detection. As such the choice of dataset is based on a practical argument. While it is true that other audio datasets might be very interesting to study, we do not believe that the choice of dataset limits the obtained insights of our work. In all of our results/tables (continual learning and OOD detection) we provide repeated experiments for all baselines and models on the audioMNIST dataset, such that the reader can get a comprehensive understanding of what to expect on this dataset. To re-emphasize, our main goal here was to provide a first demonstration of cross-domain and multi-modal continual learning and OOD detection. This aspect is typically not present in the current continual learning literature where multi-modality is not taken into account.
>
> Question 4: “For the giant Table 1, I suggest link each acronym with the reference paper.”
> Thank you for the suggestion, we will do this in the updated version.
>
> Question 5: “Can the author give me some insights, what these KL loss demonstrated in the table?“
>
> The KL loss in the table tells us about the mismatch between approximate posterior and the chosen unit Gaussian prior distribution. We do not in a strict sense use a “beta-VAE” as we also have the classification term on top of the latent space. In that sense the beta term in our case provides an additional trade-off with respect to the classifier. If the beta value is too high and too much emphasis is placed on the KL divergence, the classifier might not be able to train properly as the classes are not cleanly separated. If the beta value is too small then the obtained distribution will be too far away from a Normal distribution. This is thus different to the purely unsupervised disentanglement scenario as claimed by the original beta-VAE paper. In order to be more clear we will add a supplementary material section where we visualize 2-D latent spaces for different models for different beta values and their respective classification accuracies. This will give us the chance to provide a more in-depth discussion on the trade-off between the inevitable creation of low-density regions for the sake of class separation and thus classifier quality and the quality of the distribution alignment. The KL values in the table reflect this as one can observe that our proposed model achieves a better KLD with improved replay in contrast to unconstrained sampling, but at the same time has to balance the trade-off with respect to classifier quality. One can thus observe that the pure VAE (seen in the separate dual model approach) achieves a better KLD without the presence of the classifier.
>
>
> We will be happy to answer any additional questions and discuss further if necessary.

---

### Official Review · AnonReviewer3 · 2019-11-29
**Official Blind Review #3**

**Rating:** 6

**Review:**

Summary: This paper proposes a unified model for continual learning and aims to address the following problems:
Out-of-train-domain dataset recognition
Catastrophic forgetting
The out-of-domain or open set recognition model is not only used to detect outliers but also for sampling “representative data” of previous tasks for forward (and backward) transfer.

COMMENT(S)/QUERIES:
------------------------------

While the experiments do justice to the contributions mentioned in the paper, I believe that certain sections need clarifications and expansion (while some can be dismissed).

1. Given my limited knowledge in the literature on this topic, I would have appreciated a proper related work section.

2. I believe equation (1) could have been moved to supplementary for reference since equation (2) is the only important equation in the paper.
I don’t believe Equation (1) is really used for testing purposes. It’s just that the data is sent through the probabilistic encoder and then classified. There is no need for reconstruction of the data point.

3. It’s difficult to follow the flow of the method. Shouldn’t generative replay (algorithm 3) be placed before open set recognition of unknown and uncertain inputs (algorithm 2), since the latter is probably just used during the test, while the former affects the training procedure directly (implicit data augmentation)?

4. Major concern: It’s somewhat strange to observe that the VAE model is able to generate multiclass data so fluidly with a simple gaussian prior. This kind of challenges the belief that current generative models are unable to capture all modes perfectly. A small note about why multimodal prior was not used (which intuitively and mathematically makes more sense) and also a statement about the average time to generate multiclass multi-modal data using algorithm 3 would have been nice.

5. In section 3.2, “For our single-head expanding classifier this ensures..”. While having a single-head expanding classifier is listed as one of the important contributions, it hasn’t been given enough justice w.r.t to the implementation details. Is it like adding a completely new classifier during the training of the new task?
The entire section on hyper-parameters could’ve been moved to the supplementary if space was a major constraint but compromising on details about an important contribution only weakens the paper.

I especially like figure 2 in the experiments, where the importance of Weibull CDF outlier rejection prior is highlighted.


Typo:
incorrect opening inverted comma for the word background in the introduction section (page 1)

OVERALL COMMENT
-----------------------------

The paper combines the generative, and discriminative models into one framework for multiple important tasks. While the contributions are clear in the introduction, the presentation of the paper is somewhat too complicated at a couple of places. It does not do full justice to explaining its more vital components and some parts of the method section feel like a jigsaw puzzle, where readers are heavily expected to “figure out on their own”.
With proper presentation though, I can envision this paper contributing positively to unified frameworks in general.

Due to the above reasons. I am giving it a score of 6.


**Experience Assessment:**

I do not know much about this area.

**Review Assessment: Checking Correctness Of Derivations And Theory:**

N/A

**Review Assessment: Checking Correctness Of Experiments:**

I carefully checked the experiments.

**Review Assessment: Thoroughness In Paper Reading:**

I read the paper at least twice and used my best judgement in assessing the paper.

---

### Author Response · Authors · 2019-11-08
**Preamble for both reviewers**

Dear Reviewers,

We thank both of you for taking the time to review our work and give feedback. We would like to use this opportunity to answer your open questions and provide some clarifications.  We structure our comments as follows: we first provide a revised summary of the main elements of our paper that pertains to your questions on novelty of the paper’s contribution. This preamble is intended for both reviewers.

We then clarify technical questions and address specific points in your individual reviews that we believe may be due to misunderstanding of how our paper relates to other work in the literature.

Essence of the Contribution of the paper:

Continual learning can be viewed as sequential statistical estimation. Accumulation of bias and errors in sequential estimation is a well-known problem. Knowledge of the error characteristics of the estimated output that is derived by meta-analysis of the statistical estimator can be used to mitigate the propagation of errors in sequential estimation.  Open set outlier detection essentially uses knowledge of the probability distribution of the extreme values of sample distances from the mode. This mechanism allows to detect out-of-distribution examples during inference and to generate samples, during ‘generative replay’, that are more likely to be inliers of the sequential estimate. Use of this filtering mechanism thus prevents garbage predictions for unseen unknown classes and at the same time allows for mitigation of propagation of errors in sequential estimation.

Key Requirement: Generative replay use in continual learning should create virtual samples at any stage of the estimation that are robust with respect to errors in the estimated posterior distribution for the generative model. Prediction of unseen unknown instances should not result in a false attribution to the classifier’s known set of classes and instead be properly identified as unknown.

Key Problem: Given that a variational approximation is used for estimating the generative model, the estimate is by definition biased based on the choice of the prior. Thus the estimated pdf will have non-zero probabilities for latent vector values that do not represent samples that have been seen before. Sampling from a biased posterior results in outliers, in the sense that the samples generated can be intuitively viewed as potential samples with non-zero probabilities of occurrence.  Moreover, in a continual learning setting the posterior distribution errors accumulate over new batches of samples. In addition to this, trained neural network classifiers tend to produce false overconfident predictions even for unseen unknown classes. This is true even when epistemic uncertainty is captured through variational approximations. Conventional thresholding of metrics such as reconstruction loss or predictive entropy is not enough to correctly identify unseen unknown instances.

Solution: By flagging outliers using open set recognition, one is able to mark classifier predictions as untrustworthy for statistically different unseen unknown instances and reject samples that are due to bias in posterior distribution for the generative model.

Thus, the solution offered in the paper is a novel combination of OOD with generative modelling in a single framework. In addition to introducing natural OOD detection capability, our proposed mechanism is further complementing the generative model estimation and allows for appropriately generating virtual samples that are most likely to have been seen in the past. This is the reason why the formulation performs well.

Unfortunately, we have received the impression that both of your comments mainly address only one part of our paper’s contribution, namely the generative replay. We would appreciate it very much if you would also consider viewing our work in the context of open set recognition for outlier detection of novel data instances (see e.g. algorithm 2, section 3.4 and table 3). We understand that the presented work contains contributions that address multiple challenges. However, we believe that the introduced framework provides a natural way to address open questions in both OOD detection and continual learning at the same time.

---

### Author Response · Authors · 2019-11-11
**Uploaded revision**


Dear reviewers,

we have uploaded a revision of our paper, where we have incorporated your feedback. In the following we summarize the changes and then link them to your corresponding requests in detail below. We will be very grateful if you take some additional time to review the changes.


Short summary of changes: We have improved the presentation of our contribution and differences to the literature, improved the flow of the methods section, placed tables and figures closer to their reference point, corrected beta in the equations and have added additional remarks and a supplementary material section where the role of loss normalization and the beta term are further clarified.


Detailed individual change summary:
•	Flow:
The flow of our methods section has been rearranged. We now first introduce the model with factorization assumptions, before proceeding with the loss functions and further generative replay details.

•	Placement of figures and tables:
We have re-organized all figures and tables to be close to their first point of reference.

•	Contribution:
We have lightly extended our abstract to place more emphasis on the open set challenge tackled in our work (not visible in the abstract visible directly on open review, but inside the pdf). In addition, we have rephrased the last paragraph of the introduction section where our contribution is explicitly listed. It should now be clear that we unify class incremental continual learning with open set recognition in a single model and that this natural robustness with respect to unseen unknown examples is missing in the current continual learning literature.

•	Difference with respect to the literature/related work:
We have thoroughly thought out the possibility of adding a separate related work section. Given the strict 10-page limit, we could not realize this in a viable fashion, given the required review of both topics: continual learning and open set recognition. However, we believe that our current introductory section already provides a comprehensive overview and pointers to full length reviews. We do agree with the reviewer’s initial statement that some differences to existing literature might have been too subtle, so we have decided to include differences more explicitly (like the difference to OpenMax) where appropriate in the text. In our revised contributions list we have also extended the text accordingly. We hope that this provides readers that are less involved in the specific research topics with a better reading experience.

•	Direct references in the tables:
Unfortunately, while making the corresponding changes, we have noticed that the strict ICLR citation format of “author et. al + year” leads to significant space in the tables being occupied by references. This then shrinks the entire table and renders the results unreadable. We agree that this would have been a great idea were the citation format numerical but have decided to keep the tables as they are for improved legibility.

•	Role of beta:
We have included additional statements in the main body on the significance of the beta value, the trade-off between classification accuracy and KLD and the normalization of the loss function. To better illustrate and further explain this trade-off we have included an additional supplementary material section (B) with quantitative loss values and latent space visualization examples for different beta values.

•	Practical implementation - role of beta in conjunction with loss normalization:
We understand that we could have clarified the nuances behind the choice of beta value more explicitly and have therefore extended our explanation. Specifically, and as mentioned in the text, we normalize the loss function according to respective dimensions, i.e. input spatial dimensions for reconstruction loss, latent space dimensionality for KLD and number of classes for classification loss. This leads to a different ratio between the loss terms in contrast to a traditional beta-VAE (where reconstruction loss heavily outweighs KLD). While the results in our tables are reported in an unnormalized fashion for reference with literature values, we train with a normalized loss function. The exact value of beta is thus not comparable. In the above-mentioned appendix section, we provide further intuition on the role of the precise value of beta in practice. We would be grateful if the reviewers could consider this section.

---

### Decision · Program_Chairs · 2019-12-19

**Decision:**

Reject

**Comment:**

This paper presents a unified probabilistic approach for deep continual learning by combining generative and discriminative models into one framework that solves the following problems: catastrophic forgetting, and identifying out of distribution and open set examples. The method termed, OCDVAE in the paper achieves closer or better to SOTA results in different evaluation tasks.

The reviewers had several concerns about the presentation of the paper and some errors in the equations, all of which seem to have been fixed in the latest upload made by the authors. Blind review #3 was delayed as the original reviewer refused to review the paper and this review was then obtained by someone else after the new upload of the paper, so this review looks at the new version of the paper. I would recommend the authors to incorporate suggestions provided by reviewer #3 in the final version of the paper including expanding on the related work section.

However, as of now I recommend to reject the paper.